# Distribution Transformers: Fast Approximate Bayesian Inference With On-The-Fly Prior Adaptation

**George Whittle** [1,2]  **Juliusz Ziomek** [1]  **Jacob Rawling** [2]  **Maike A. Osborne** [1,2]

## Abstract

While Bayesian inference provides a principled framework for reasoning under uncertainty, its widespread adoption is limited by the intractability of exact posterior computation, necessitating the use of approximate inference. However, existing methods are often computationally expensive, or demand costly retraining when priors change, limiting their utility, particularly in sequential inference problems such as real-time sensor fusion. To address these challenges, we introduce the Distribution Transformer—a novel architecture that can learn arbitrary distribution-to-distribution mappings. Our method can be trained to map a prior to the corresponding posterior, conditioned on some dataset—thus performing approximate Bayesian inference. Our novel architecture represents a prior distribution as a (universally-approximating) Gaussian Mixture Model (GMM), and transforms it into a GMM representation of the posterior. The components of the GMM attend to each other via self-attention, and to the data-points via cross-attention. We demonstrate that Distribution Transformers both maintain flexibility to vary the prior, and significantly reduces computation times—from minutes to milliseconds—while achieving expected log-likelihood performance on par with or superior to existing approximate inference methods across tasks such as sequential inference, quantum system parameter inference, and Gaussian Process predictive posterior inference with hyperpriors.

[1]Department of Engineering Science, University of Oxford, Oxford, United Kingdom [2]Mind Foundry Ltd, Oxford, United Kingdom. Correspondence to: George Whittle <george.whittle@reuben.ox.ac.uk>.

*Proceedings of the 43rd International Conference on Machine Learning*, Seoul, South Korea. PMLR 306, 2026. Copyright 2026 by the author(s).

## 1. Introduction

Bayesian inference provides a principled route to uncertainty quantification and the incorporation of prior knowledge. In practice, however, repeatedly solving inference problems (e.g., across datasets, hyperparameters, or environments) is computationally expensive. *Amortised Bayesian inference (ABI)* addresses this by learning a mapping from observations to posterior approximations, yielding fast test-time inference. Recent transformer-based ABI methods have shown impressive single-pass inference for small-data regimes (Müller et al., 2021; Hollmann et al., 2022; 2025).

Yet two limitations persist. First, most ABI models *fix the prior* during training; changing the prior at test time typically requires retraining or fine-tuning. Second, methods that offer some prior flexibility usually *do not preserve family structure* between prior and posterior, hindering sequential composition (filtering/smoothing), where the posterior must become the next-step prior. Complementary lines in amortised SBI (Cranmer et al., 2020) and sensitivity-aware amortisation (Elsemüller et al., 2024) underscore the need for flexible priors, but do not maintain the at-least approximate conjugacy needed for recursive updates.

We introduce *Distribution Transformers (DTs)*, a transformer-based architecture that **(i)** performs single-pass amortised inference, **(ii)** amortises inference *across a family of priors at test time*, and **(iii)** approximates *conjugacy* between prior and posterior, enabling clean sequential composition when needed. Concretely, DTs embed priors as a tokenised sequence (representing a Gaussian Mixture Model, or GMM, approximation), condition on observations with a permutation-equivariant decoder, and output a posterior in the *same* family. Training spans a distribution over priors, enabling prior-amortisation.

Concretely, we make the following contributions:

- A unified framework for *prior-flexible*, approximately *conjugate* amortised inference, bridging the gap between ABI and sequential inference.

- A practical transformer architecture that tokenises distributions and performs posterior updates within the same parametric family. The latter of these properties

is unique even amongst prior-flexible amortised inference methods, and is essential for sequential inference applications.

- Empirical results on both *static* benchmarks (matching or exceeding PFN/TabPFN- and VI-style baselines) and *sequential* tasks (where no other amortised method can be applied), demonstrating accuracy and speed.

## 1.1. Related Work

**Amortised Bayesian Inference.** Amortisation has emerged as a key paradigm to accelerate Bayesian inference across repeated tasks by transferring expensive optimisation costs to an offline training phase. Classical amortised variational inference (AVI) approaches (Kingma & Welling, 2013; Ganguly et al., 2023) learn inference networks that map directly from observations to approximate posteriors, but typically rely on restrictive variational families and do not support flexible prior specification. Recent extensions have explored amortisation in simulation-based inference (SBI), leveraging normalising flows (Rezende & Mohamed, 2015; Papamakarios et al., 2017), neural ratio estimation (Greenberg et al., 2019; Miller et al., 2022), and hybrid MCMC-amortised approaches (Salimans et al., 2015; Gabrié et al., 2022), but these remain tied to fixed priors at training time. Methods amortising inference across priors *as well as tasks*, termed prior-flexible amortised inference methods, have only recently been proposed, for example prior-amortized neural posterior estimation for reflectometry inversion (Starostin et al., 2025), and sensitivity-aware amortised inference (Elsemüller et al., 2024), highlighting both the importance and nascent state of this line of work. Relatedly, BayesFlow provides a framework based on invertible neural networks and tooling for practical workflows (Radev et al., 2020).

**Transformers for Bayesian Inference.** Transformers have recently shown promise as amortised inference engines. Prior-Fitted Networks (PFNs) (Müller et al., 2021) demonstrated that transformers can approximate posterior distributions in a single forward pass, amortising inference over datasets. However, PFNs assume a fixed prior, and their Riemannian output distribution struggles with smooth or heavy-tailed posteriors. Follow-up work extended PFNs to tabular data (TabPFN) (Hollmann et al., 2022), scaling to larger contexts and small-data regimes with improved accuracy (Hollmann et al., 2025), and to time-series forecasting (Hoo et al., 2024). Nonetheless, these methods remain restricted to fixed priors[1]. In parallel, amortised in-context Bayesian inference methods (Mittal et al., 2025;

Reuter et al., 2025) investigate whether transformers can learn posterior inference directly from prompts, but still lack mechanisms for prior adaptation.

**Meta-Learning and Neural Processes.** Our work connects to neural processes (Garnelo et al., 2018a;b), which condition on context sets to predict function values, and their transformer-based extensions (Kim et al., 2019; Nguyen & Grover, 2022). While these frameworks amortise inference across tasks, they typically model predictive distributions over data rather than explicit posteriors over latent variables. ACE (Chang et al., 2024) generalised this direction to incorporate latents and flexible priors, and is thus complementary to our work. Other amortised meta-learning approaches (Wu et al., 2020; Iakovleva et al., 2020) have proposed shared amortised inference networks across tasks, but again without explicit prior adaptation.

**Simulation-Based Inference.** SBI methods (Cranmer et al., 2020; Lueckmann et al., 2017; Papamakarios et al., 2019) provide amortised posterior approximations for complex simulators, with recent advances leveraging transformers and diffusion models (Gloeckler et al., 2024; Sharrock et al., 2022; Wildberger et al., 2023). These works are highly expressive and achieve state-of-the-art inference in scientific applications, yet typically assume a fixed prior distribution or limited parametric families. This lack of efficient prior flexibility limits their applicability in scenarios requiring frequent prior updates, such as sensitivity analysis or robust sequential decision-making.

**Sequential Inference.** Classical Bayesian filtering methods–such as the Kalman filter (Kalman, 1960), unscented Kalman filter (Julier & Uhlmann, 1997), and particle filters (Doucet et al., 2001; Wills & Schön, 2023)–remain the dominant approaches to sequential inference. While computationally efficient, they are either constrained to Gaussian assumptions or expensive particle-based representations, and provide no general amortisation across tasks or priors. Despite the centrality of sequential inference in real-world applications, modern amortised inference frameworks (PFNs, TabPFN, ACE) have largely neglected this dimension.

**Positioning.** In summary, prior work has established the effectiveness of amortisation in approximate Bayesian inference, and the suitability of transformers for fast, in-context posterior approximation. However, *prior-flexible amortisation*, particularly in the context of sequential inference, remains largely unaddressed. Our work introduces Distribution Transformers as a principled solution, combining (i) the expressive universality of Gaussian mixtures, (ii) transformer-based amortisation across priors, and (iii) applicability to sequential Bayesian filtering—providing ca-

---

[1] A rudimentary form of prior-adaptation can be carried out by providing prior parameters as additional observations, but this is restrictive and a side-effect, not a directly-intended feature.

pabilities unmatched by existing PFN, TabPFN, ACE, or simulation-based inference approaches.

## 2. Preliminaries

### 2.1. Transformers

The transformer architecture (Vaswani et al., 2017) has revolutionised deep learning, achieving state-of-the-art performance across domains including language modelling (Brown et al., 2020), computer vision (Dosovitskiy et al., 2020), and Bayesian inference (Müller et al., 2021). At their core, transformers learn mappings between sequences of tokens through the attention mechanism (Bahdanau et al., 2014), which enables parallelised information flow between sequence elements. This mechanism, combined with token-wise MLP layers, creates a parameter-efficient architecture capable of processing sequence elements in parallel.

Two theoretical properties of transformers are central to our work. First, transformers are universal sequence-to-sequence approximators (Yun et al., 2019), capable of learning arbitrary mappings between sequences. Second, in the absence of positional encodings, transformers are permutation equivariant with respect to the input sequence—a property we exploit in Section 3.

Our method specifically employs the transformer decoder architecture. This architecture extends the base transformer by incorporating global cross-attention layers, allowing each token in the input sequence to attend to a separate context sequence. This cross-attention mechanism provides a natural framework for conditioning sequence transformations on observed data.

### 2.2. Gaussian Mixture Models

Gaussian Mixture Models (GMMs) are flexible probability distributions whose density is a weighted sum of Gaussian components, i.e. $q(x) = \sum_i w_i \mathcal{N}(x; \mu_i, \Sigma_i)$ where $\mathcal{N}(x; \mu, \Sigma)$ is a Gaussian density over $x, w_i \in [0, 1]$, $\sum_i w_i = 1$, $\mu_i \in \mathbb{R}^n$, and $\Sigma_i \in \mathbb{S}^n_{++}$, illustrated in Figure 1. While GMMs are widely used in clustering (Dempster et al., 1977), and latent variable modelling (Bishop, 2006), we focus on their role as universal approximators of smooth probability distributions (Goodfellow et al., 2016; Calcaterra & Boldt, 2008)—a property we exploit in Section 3. This universality extends to distributions on compact domains under appropriate change of measure, also illustrated in Figure 1. While the idea of using GMMs for function approximation is not new to deep learning (for example Bishop 1994 proposes the use of a GMM to model uncertainty in the output of a neural network), the idea of operating end-to-end on a distribution represented as a GMM is novel. A key property of GMMs is their natural representation as an unordered sequence of component parameters. Specif-

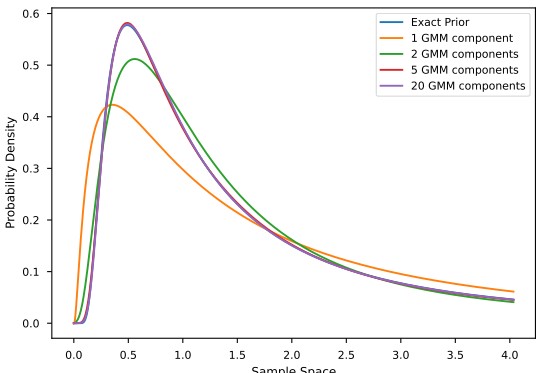

*Figure 1.* Various log-warped GMM approximation to an inverse-gamma prior distributions. Note that even with only five GMM components, the approximation is visually almost indistinguishable from the target distribution. This is true for many frequently encountered distributions in Bayesian inference.

ically, a $k$-component GMM over $\mathbb{R}^n$ is parametrised by $\boldsymbol{\theta} = \{(w_i, \boldsymbol{\mu}_i, \boldsymbol{\Sigma}_i)\}_{i=1}^k$. This representation is permutation invariant—the ordering of components does not affect the resulting distribution. Fitting a GMM to a given probability distribution is non-trivial, with methods such as expectation maximisation and variational approaches suffering from similar problems to their approximate inference counterparts. We will show that DTs naturally provide a mapping from the parameters of a given distribution to an approximating GMM without introduction of additional model parameters relative to a model which decodes the posterior only.

## 3. Distribution Transformers

Given a prior distribution $p(x \mid \phi)$ from a parametric family with parameters $\phi \in \Phi$ and observations $z \in \mathcal{Z}$ governed by likelihood $p(z \mid x)$, Bayesian inference aims to compute the posterior $p(x \mid z, \phi)$. Amortised approximate Bayesian inference reframes this as learning a mapping $\Phi \times \mathcal{Z} \to \mathcal{Q}$, where $\mathcal{Q}$ is a space of approximate posteriors. We introduce the **Distribution Transformer (DT)**, a transformer-based architecture that directly maps priors and observations to posteriors. A core challenge in Bayesian inference is representing arbitrary probability distributions in a form suitable for neural networks. **Our first key innovation** is to represent all distributions as Gaussian Mixture Models (GMMs), which approximate any continuous density arbitrarily well. **Our second key innovation** is a transformer architecture that processes these mixtures as unordered sequences, preserving probabilistic structure while enabling expressive, scalable inference. Figure 2 illustrates the DT architecture in detail. Conceptually, the DT architecture can be broken into four parts: the prior embedding, observation embeddings, transformer decoder and GMM unembedding.

Obtaining a GMM representation of an arbitrary prior is

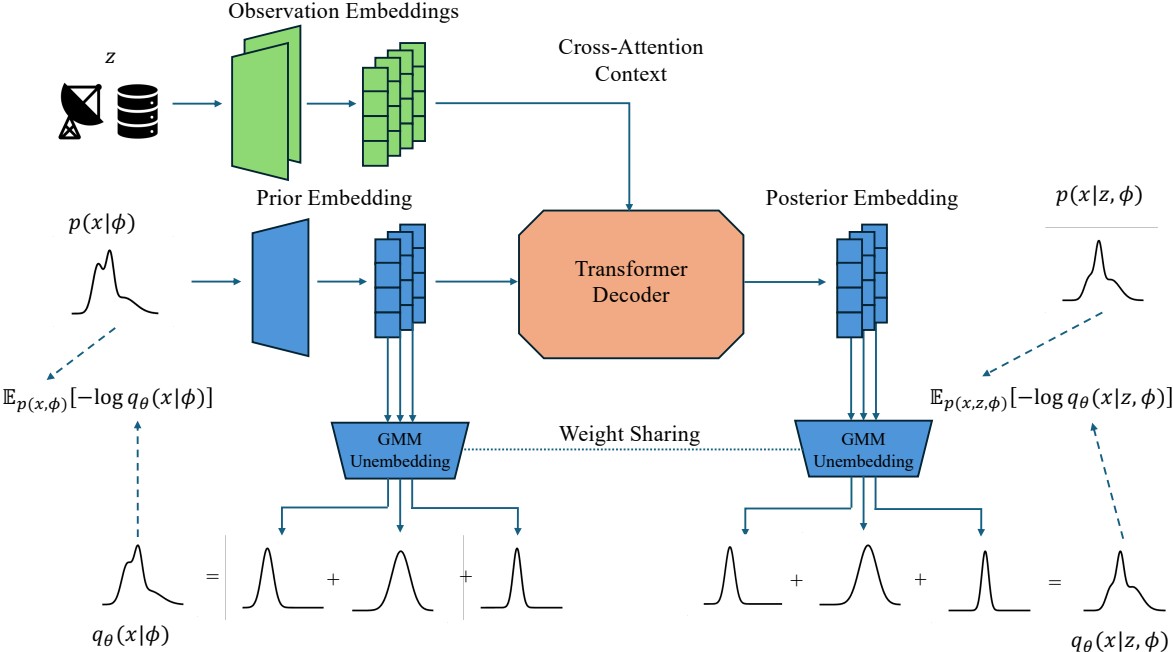

*Figure 2.* Architecture diagram for a distribution transformer. Observations, e.g. from a dataset or a sensor measurement, are transformed to a set of tokens in the latent space via a distinct learnable embedding for each datasource. Priors are represented as a set of embedded GMM components in the latent space via a learnable embedding acting on their parameters. The distribution transformer itself, a transformer decoder, learns to map the prior to the posterior in the latent space, incorporating information from the embedded observations via cross attention. A learnable unembedding acts token-wise on both the prior and posterior latent GMM representations to give a GMM approximation for both the prior and posterior distributions, with which we estimate our loss function $\ell'_\theta$.

nontrivial. Inspired by (Bishop, 1994), we introduce a learnable embedding network that maps a prior's parameters to a length-$k$ unordered sequence in the transformer's latent space, representing an embedding of a $k$-component GMM approximation to the prior.

Observations $z$ vary widely (e.g., sensor readings, datasets). We use learnable embeddings tailored to different data sources, and embed datasets as sequences of data-label pairs embedded token-wise by a single embedding model. If a predictive posterior is needed, the query point is embedded separately. Once embedded, observations are combined as a unified latent sequence suitable for input to a transformer.

The transformer decoder maps the latent GMM prior representation to a posterior representation, conditioned on the observations via global cross-attention. We omit positional encodings to preserve permutation equivariance, aligning with the permutation invariance of the GMM representation.

A GMM posterior approximation is then obtained via a learnable unembedding that acts component-wise on the posterior latent GMM unordered sequence, producing logits and normal densities. A cross-sequence softmax converts logits into component weights, and the approximating GMM can be constructed through summation of these components, achieving end-to-end permutation invariance of the architecture, as required.

Now that we have an architecture capable of mapping between distributions, we propose a sample-based training scheme with which to train our architecture to perform Bayesian inference. We must first introduce the concept of meta-priors $p(\phi)$—priors over priors representing the expected distribution of priors encountered by the algorithm. The only constraint on these meta-priors is that they can be sampled from, and can otherwise be quite complicated. For instance, in vehicle tracking, a meta-prior could constrain Gaussian means to a city's road network while shaping covariance to reflect realistic uncertainties. Using this meta-prior, we can specify the joint distribution $p(\phi, x, z)$ hierarchically as $p(\phi)p(x \mid \phi)p(z \mid x)$. We may also specify a mapping $f(\cdot)$ from the sample space of interest to the sample space of the approximating GMM $\mathbb{R}^n$, for example to account for priors with finite support, essentially specifying a change of measure ensuring the probabilistic properties of the approximation are maintained. In this case, we denote the GMM itself as $q_\theta(f(x) \mid z, \phi)$, inducing the warped GMM $q_\theta(x \mid z, \phi) \approx p(x \mid z, \phi)$ under change of measure.

Outlined in Algorithm 1, DTs are trained to minimise $\ell'_\theta$. Using meta-priors and a sample-space transform $f$, we extend the loss function proposed by Müller et al. (2021) another meta-level, defined as $\ell_\theta = \mathbb{E}_{p(\phi, x, z)} \left[ -\log q_\theta(f(x) \mid z, \phi) \right]$. We show that this is equivalent to direct minimisation of the KL-Divergence between the true posterior $p(x \mid z, \phi)$ and

**Algorithm 1** Training a Distribution Transformer

**Inputs:** A joint distribution $p(\phi, x, z)$ over priors, latent variables, and observations; the number of training iterations $m$; the batch size $b$; the number of GMM components $k$; and a mapping $f$ from the sample space of interest to the sample space of the GMM.

**Output:** A mapping from $\phi$ and $z$ to warped GMMs $q_\theta(x \mid \phi)$ and $q_\theta(x \mid z, \phi)$ approximating the prior $p(x \mid \phi)$ and posterior $p(x \mid z, \phi)$ respectively.

**for** $i = 1$ **to** $m$ **do**

    Sample $\phi_i$, $x_i$ and $z_i$ from $p(\phi, x, z)$ for $i = 1 : b$;

    Estimate loss $\hat{\ell}'_\theta = -\sum_{i=1}^{b} \log q_\theta(f(x_i) \mid \phi_i) + \log q_\theta(f(x_i) \mid z_i, \phi_i)$;

    Update DT parameters with gradient descent on $\nabla_\theta \hat{\ell}'_\theta$;

**end for**

the GMM approximation $q_\theta(x \mid z, \phi)$:

**Proposition 3.1.** *The proposed loss $l_\theta$ is equal to the expected KL-Divergence $\mathbb{E}_{p(\phi,z)}\left[KL\left[p \mid\mid q_\theta\right]\right]$ between $p(\cdot \mid z, \phi)$ and $q_\theta(\cdot \mid z, \phi)$ up to an additive constant.*

A proof of Proposition 3.1 can be found in Appendix A.

The loss $l_\theta$ can be estimated using samples from the joint distribution $p(\phi, x, z)$, alleviating any need to directly access or sample from the posterior density.

Finally, DTs can jointly approximate priors and posteriors *without additional model parameters*. Applying the unembedding to the prior sequence yields a GMM approximation $q_\theta(x \mid \phi)$, extending DTs to mappings $\Phi \times \mathcal{Z} \to \Theta \times \Theta$. To ensure consistency and a shared latent space pre- and post-conditioning, we introduce a prior loss:

$$\ell_\theta^{\text{prior}} = \mathbb{E}_{p(\phi,x)}\left[-\log q_\theta(x \mid \phi)\right],$$

leading to the combined objective: $\ell'_\theta = \ell_\theta^{\text{prior}} + \ell_\theta$ .

The prior loss term acts as a regulariser relative to the primary task, delivering modest performance gains, but is essential for achieving latent space conjugacy.

## 4. Empirical Studies

We study the behaviour of our method in three settings: approximation of a tractable posterior, approximation of intractable posteriors, and a real-world sensor fusion problem posed as sequential inference. In the former two settings, we benchmark our method against SVI (Hoffman et al., 2013), implemented in PyTorch (Paszke et al., 2019) with GPU parallelisation, and PFNs (Müller et al., 2021), also implemented in PyTorch, fitting a Riemann distribution with the same number of model outputs as our DT. Our code is

*Table 1.* Results for Experiment 4.1 (**best**), tested on and timed over 1000 sampled unseen problems. Expected KL-Divergences with the true posterior (along with 95% confidence intervals) are given for the narrow (above) and wide (below) meta-priors. DT-$k$ refers to a $k$-component Distribution Transformer, and PFN-$n$ refers to a PFN equipped with an $n$-bucket Riemann distribution. Note first that both variants of our proposal achieve better posterior KL-Divergences than an equivalent PFN and SVI for both meta-prior settings, while performing inference orders of magnitude faster than SVI. This difference is particularly apparent for the wide meta-prior, where the PFN fails to fit the posterior entirely.

| METHOD | KL-DIVERGENCE | INFERENCE TIME PER 1000 PROBLEMS (S) |
|---|---|---|
| SVI | 0.0425±0.0003
0.0558±0.0016 | 148 |
| PFN-15 | 0.517±1.009*
331.5±646.6* | **0.003** |
| PFN-5000 | 0.0038±0.0789
0.2935±0.0237 | **0.003** |
| TABPFNv2 | 0.0112±0.0013
0.1513±0.0168 | 1.52 |
| ACE-5 | 0.0094±0.0000
0.0048±0.0014 | 0.037 |
| DT-2 | 0.0044±0.0001
0.0058±0.0002 | 0.014 |
| DT-5 | **0.0004±0.0000**
**0.0003±0.0000** | 0.016 |

available on GitHub. [2] In the absence of a fixed prior, we train PFNs using the same sampling scheme as our method, effectively marginalising out the meta-prior, leaving a less informative prior. We expect PFNs to perform well when the meta-prior is narrow and poorly when wide, as the effective prior used by the PFN is the marginalisation of the prior family with respect to the meta-prior and so is close to the true prior only in the former case. For the latter experiment, we benchmark against the widely used extended Kalman filter (EKF) where possible, and a particle filter (PF), again with GPU implementation (Simon, 2006; Doucet et al., 2001). For the PF, we obtain a density via Gaussian kernel density estimation.

### 4.1. Analytical Verification Study

In specific cases, where the adopted prior is of a conjugate family to the likelihood, the posterior is tractable. We first verify that our approach indeed performs approximate inference, for the case of an inverse-gamma prior and normal-variance likelihood. We choose a meta-prior consisting of independent inverse-gamma distributions over the rate and shape parameters, and test our approach on both narrow and wide meta-prior settings.

---

[2]https://github.com/GWhittle110/distribution-transformers

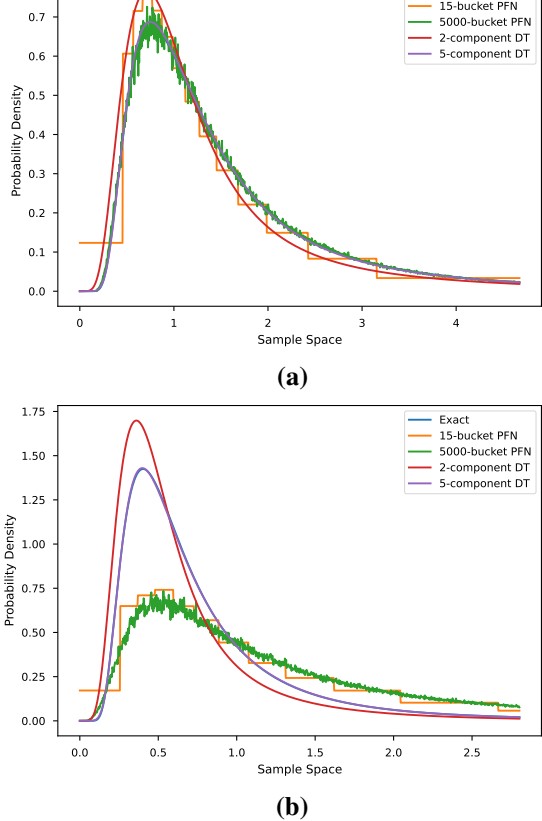

**(a)**

**(b)**

*Figure 3.* Ground truth, PFN, and 2 and 5 component DT posterior densities for an inverse-gamma prior with an (a) narrow and (b) wide meta-prior. Both variants of the DT fit the true posterior well, in both cases with the 5 component DT almost indistinguishable from the ground truth. The PFN's shape is correct in both cases, and fits the ground truth correctly (up to the limits of the Riemann distribution) for the narrow meta-prior, but as expected completely fails to fit the ground truth for the wide meta-prior, given the lack of prior. In any case, for a given number of model outputs the DT provides a much tighter fit to the ground distribution.

Figure 3 demonstrates that for both wide and narrow meta-priors, the Distribution Transformer does indeed learn to perform Bayesian inference and provides excellent approximations to the posterior, even under change of prior. Figure 1 shows the GMM approximations provided by the DT for this study, demonstrating that even with no additional model parameters a high-quality mapping to a GMM approximation of the prior is achieved. It is clear that while PFNs perform as well as the Riemann distribution allows for the narrow meta-prior case, as expected in the wide case they fail to fit the posterior at all. This is further demonstrated in Table 1, which shows that not only do we achieve speeds close to that of PFNs, which are slightly faster due to the lighter-weight architecture, but even significantly outperform the much slower SVI in terms of posterior KL-Divergence. These performance gains can be attributed to the expressive GMM adopted by our method, and the efficient transformer-based

architecture. TabPFNv2, while a much larger model than any other baseline, is also unable to improve much on top of standard PFNs. ACE achieves second best performance, losing only to DTs. It thus appears that all methods utilising GMMs surpass those with Riemannian predictives. We attribute the performance gains of DTs over ACE primarily to a more flexible embedding setup, which our architecture allows for.

An interesting observation, is the extremely high uncertainty in the estimate for the PFN posterior KL-Divergence, marked *. This can be attributed to a failing of the Riemann distribution in this setting—the half-Gaussian tail adopted by the Riemann distribution has variance fitted to the (marginal) prior, meaning for certain observations (or priors), the true posterior has significant probability mass in the right tail which the Riemann distribution cannot express, leading to a skewed distribution for the expected KL-Divergence with a misleading 95%-confidence interval.

### 4.2. Posterior Approximation Studies

We now move our attention to problems where the posterior is intractable, as is more often the case.

#### 4.2.1. GAUSSIAN PROCESS JOINT PREDICTIVE POSTERIOR AND HYPERPOSTERIOR

When modelling data with a Gaussian Process, it is common to assign priors to hyerparameters, known as hyperpriors. These hyperpriors render the predictive posterior intractable, and moreover, the posterior for the hyperparameters, or hyperposterior, is also intractable. Existing techniques tackle these distributions separately, and are plagued by the aforementioned issues. We now demonstrate that our method can quickly perform approximate inference jointly over both the predictive posterior and hyperposterior.

In Table 2 we show that we outperform existing methods on a more challenging 5-dimensional input problem in terms of NLL for both PPD and hyperposterior, as expected, while also being the fastest method in terms of runtime. In Figure 4 we show an example PPD, with DTs clearly closer to the oracle PPD equipped with the true lengthscale.

#### 4.2.2. QUANTUM SYSTEM PARAMETER INFERENCE

An interesting example of an inference problem involving genuine randomness with real-world implications is parameter inference for a quantum system. For this experiment, we infer the unknown parameter $\Delta$ for a two-level quantum system with Hamiltonian $H = \Delta\sigma_x + (1-\Delta)\sigma_z$, where $\sigma_x$ and $\sigma_z$ are the Pauli X and Z matrices respectively. We model observations as 10 independent experiment runs, subject to uncertainty in initial state preparation and measurement times, and generated via a GPU-implemented simulation.

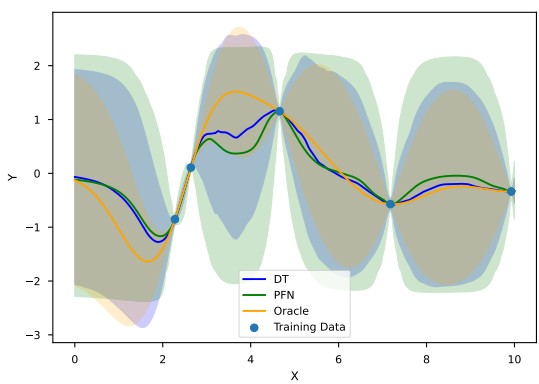

*Figure 4.* Example plot for the 1-dimensional input GP predictive experiment with hyperpriors, using 10 GMM components. Here we put an InverseGamma(1,2) prior on the lengthscale. We plot our model's predictive posterior in blue and PFNs in green. In orange, we show the oracle that is the exact GP fit with the true lengthscale value (which is unobserved for the other methods). We see that PFNs overestimate the confidence intervals due to the fact that they do not take the prior, particularly that of the lengthscale, into account.The Riemann distribution of the PFN uses 30 buckets, matching the number of outputs of the DT.

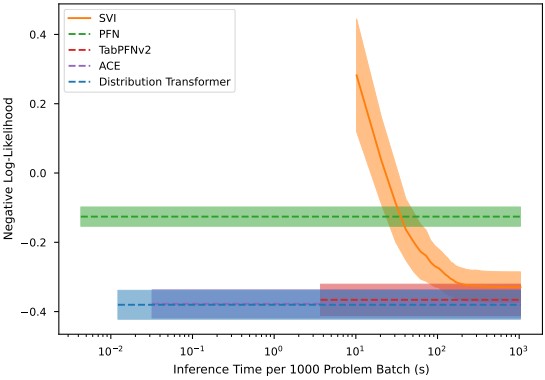

*Figure 5.* Expected negative log-likelihood against batch inference time. Note that even given orders of magnitude more computation time, SVI cannot match the performance of our method, again demonstrating the power of our GMM approximation. Note that this problem is particularly challenging for VI, as the likelihood must be marginalised with respect to the uncertainty in the initial state and measurement time, which is not tractable and must be estimated stochastically, increasing the time per iteration.

Figure 5 illustrates the power of our approach, achieving better expected log-likelihood performance (and therefore a smaller expected KL-Divergence with the true posterior, indicating a better fit), than PFNs and SVI and matching performance of TabPFNv2 and ACE, while being faster. This is a problem setting where variable priors are important, as the prior sensitivity analysis is typically necessary. To enable such an analysis, fast inference is crucial. This further illustrates the practical advantage of our approach, as we explicitly target these problems.

*Table 2.* Expected NLL for the marginal PPD and the marginal hyperposterior (denoted Hyper), and inference time per 1000 problems (denoted Runtime), for Experiment 4.2.1. VI is not evaluated for the PPD, as is convention. As expected, we outperform in both categories. Note that PFNs suffer the same issue with the hyperposterior NLL confidence here as in Experiment 4.1. The usually-accurate MCMC also underperforms here, as certain meta-priors give rise to instability and non-convergence, and thus inaccuracies.

| METHOD | EXPECTED NLL | | INFERENCE |
| | PPD | HYPER | TIME (S) |
|---|---|---|---|
| VI | ✗ | $0.39 \pm 0.05$ | 123 |
| MCMC | ✗ | $2.27 \pm 0.04$ | 27 |
| PFN | $0.90 \pm 0.02$ | $1.53 \pm 0.01^*$ | **9.2** |
| TABPFNv2 | $1.08 \pm 0.02$ | $0.37 \pm 0.02$ | 28.5 |
| ACE | $1.08 \pm 0.02$ | $0.35 \pm 0.02$ | 11.7 |
| DT | $\mathbf{0.81 \pm 0.02}$ | $\mathbf{0.31 \pm 0.02}$ | 9.5 |

### 4.3. Sequential Inference Studies

The primary advantage of our approach over other prior-flexible amortised inference methods is *conjugacy*: modelling both the prior and posterior as a multivariate GMM. This property unlocks DT's unique capability to be applied in *sequential*, Bayesian filtering-like inference settings, where a previous step's posterior is propagated to the next step's prior. One may reasonably ask why the previously discussed methods should not be used here by sequentially appending observations as they arrive; obvious difficulties in incorporating knowledge of the system dynamics aside, this approach will cause inference time to scale at least linearly with sequence length $T$ (if not $\mathcal{O}(T^2)$, as is the case when they are implemented with a vanilla transformer), while by enabling the Bayesian filtering inference paradigm our method achieves inference time constant-in-$T$.

For baselines in this setting, we turn to standard algorithms for real-time sequential inference: the Extended Kalman Filter (EKF) and Particle Filter (PF) (Simon, 2006; Doucet et al., 2001). The EKF linearises system dynamics and observation models, and approximates all sources of uncertainty as additive Gaussian. However, these assumptions are rarely reflected in reality. The PF, otherwise known as Sequential Monte Carlo, propagates a cloud of particles representing the distribution over latents and uses some variant of weighted resampling to condition on observations. While asymptotically exact, PFs are computationally intensive and suffer heavily from the curse of dimensionality. Our approach is well-suited to this problem setting, as time spent training is almost irrelevant, and fast, conjugate inference with variable priors is the priority.

We evaluate DTs on two real-world problem settings: firstly, a sensor fusion problem consisting of 2-dimensional linear dynamics, modelled as a 4-dimensional state space, with indirect measurements of displacement provided by

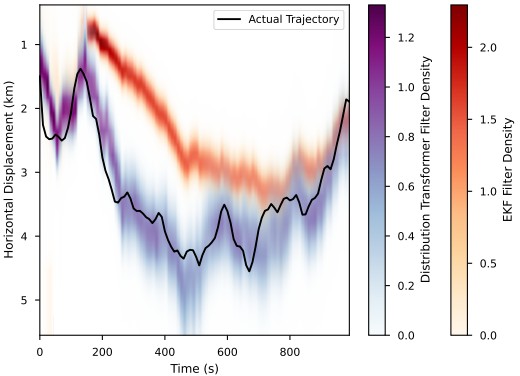

*Figure 6.* Marginal filter densities of horizontal displacement for both an EKF and a 4-component Distribution Transformer. Clearly, the DT tracks the true trajectory much more accurately, which is reflected in the superior expected NLL reported in Table 3. Note that the DT generally has a higher uncertainty than the EKF, demonstrating proper handling of the complex uncertainty structure of the observations.

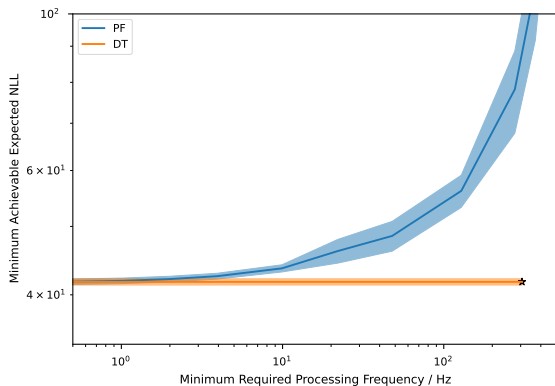

*Figure 7.* Minimum Achievable Expected NLL against Minimum Required Processing Frequency for PFs and DTs on Experiment 4.3.2. Even in this higher-dimensional setting, DTs achieve expected NLL performance only matchable by prohibitively large PFs, at a speed only achievable by severely performance-compromised PFs.

### 4.3.2. Factor-Structure Stochastic Volatility

Figure 7 clearly demonstrates that even in this challenging setting, where information is so sparse such that even the limiting PF Expected NLL (our assumed ground truth) is very high, DTs perform competitively with PFs requiring three orders of magnitude more computation time per iteration. This clearly demonstrates the potential of our proposal on a challenging, realistic scenario where few baselines exist, especially within the amortised inference cohort.

## 5. Conclusion and Limitations

In summary, Distribution Transformers (DTs) introduce a powerful framework for approximate Bayesian inference by combining universally-approximating GMMs with Transformer architectures. Our empirical results demonstrate that DTs not only achieve superior inference accuracy and speed compared to existing methods, but also enable dynamic prior updates without retraining *while maintaining conjugacy*—a unique capability in the field. Furthermore, DTs show competitive inference performance in terms of expected posterior NLL on challenging tasks, while maintaining the computational efficiency needed for practical applications, particularly in sequential inference settings.

We also acknowledge the limitations of our approach. Being a prior-adaptive method, training must cover a higher-dimensional space, increasing offline training cost relative to fixed-prior baselines (see Table 8 for concrete figures). Prior-adaptive methods also require a reasonably well-specified meta-prior, although preliminary results indicate some robustness to meta-prior misspecification (see Appendix C.2). Being based on multivariate GMMs, DTs inherit their weaknesses in high-dimensional problems (although we note that

*Table 3.* Expected NLL and iteration time (prediction, update and distribution generation) for EKF, PF, and DT. Note that our method vastly outperforms the EKF in terms of NLL with a small cost in iteration time, while almost matching the close to ground-truth PF's NLL but achieving close to 50× speedup.

| Method | Expected NLL | Iteration Time for 100 Series Batch (s) |
|---|---|---|
| EKF | 95.9±4.40 | **0.010** |
| PF | **-0.244±0.047** | 0.818 |
| DT | **-0.197±0.040** | 0.017 |

two independent, non-linear, non-Gaussian, sensors. Secondly, a 10-dimensional factor-structure stochastic volatility model, with factor log-volatilities modelled by independent Ornstein-Uhlenbeck processes, and 30 observations per timestep of known, but random, factor loadings corrupted by idiosyncratic variance. The latter of these is extremely challenging, requiring a large number of particles for the PF to converge, and downright prohibiting use of the EKF as the likelihood's mean is independent of the latents.

### 4.3.1. Bayesian Sensor Fusion

Figure 6 clearly demonstrates that our approach tracks the true state well, while the EKF fails to track the true state at all. Table 3 confirms this, and shows that our approach sacrifices little in terms of iteration time, which upper bounds the frequency at which observations can be processed in real time, compared to the EKF. As expected, the close-to-ground truth PF achieves marginally better NLL than our method, at the expense of a significant slowdown.

many real-world high-dimensional problems may still be tackled effectively using GMMs with a modest number of components, for example see Luxenberg & Boyd (2024)): computational complexity is quadratic in the number of GMM components (via self-attention) and quadratic in the underlying latent variable dimension (via full-covariance decoding per token), with the latter being the dominant memory cost at scale. We note that in our most challenging experiment (Section 4.3.2), DTs continued to process series in parallel at full batch size, whereas the PF baseline required serial processing and a moment-matched substitute for the Gaussian KDE at large particle counts; nevertheless, practitioners deploying DTs at substantially higher latent dimensions should consider memory requirements of full-covariance decoding as a limiting factor. Hyperparameters were chosen by limited manual tuning; we did not observe strong sensitivity to these choices, but a more systematic study is left to future work. Finally, like any approximate inference method, use in sequential settings may accumulate error over recursions, though we find this remains small out to moderate depths, which we ablate in Appendix C.5.

## Acknowledgements

This work was supported by the Engineering and Physical Sciences Research Council (EPSRC) under grant EP/W524311/1. This research was conducted in part during employment of George Whittle at Mind Foundry Ltd. We thank Professor Natalia Ares (Department of Engineering Science, University of Oxford) for valuable discussions and feedback throughout the development of this work.

## Impact Statement

This paper presents work whose goal is to advance the field of Machine Learning. There are many potential societal consequences of our work, none which we feel must be specifically highlighted here.

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

## A. Proof of Proposition 3.1

*Proof.* This equality can shown with a simple derivation:

$$
\begin{aligned}
\ell_\theta &= \mathbb{E}_{p(\phi,x,z)} \left[ -\log q_\theta(f(x) \,|\, z, \phi) \right] \\
&= \mathbb{E}_{p(\phi,z)} \left[ \mathbb{E}_{p(x \,|\, \phi,z)} \left[ -\log q_\theta(f(x) \,|\, z, \phi) \right] \right] \\
&= \mathbb{E}_{p(\phi,z)} \left[ \mathbb{E}_{p(x \,|\, \phi,z)} \left[ -\log q_\theta(x \,|\, z, \phi) + \log |\det J_f(x)| \right] \right] \\
&= \mathbb{E}_{p(\phi,z)} \left[ \mathbb{E}_{p(x \,|\, \phi,z)} \left[ \log p(x \,|\, z, \phi) - \log p(x \,|\, z, \phi) - \log q_\theta(x \,|\, z, \phi) + \log |\det J_f(x)| \right] \right] \\
&= \mathbb{E}_{p(\phi,z)} \left[ \mathbb{E}_{p(x \,|\, \phi,z)} \left[ \log \frac{p(x \,|\, z, \phi)}{q_\theta(x \,|\, z, \phi)} \right] \right] + \mathbb{E}_{p(\phi,z)} \left[ \mathbb{E}_{p(x \,|\, \phi,z)} \left[ \log |\det J_f(x)| - \log p(x \,|\, z, \phi) \right] \right] \\
&= \mathbb{E}_{p(\phi,z)} \left[ \mathrm{KL} \left[ p(x \,|\, z, \phi) \,||\, q_\theta(x \,|\, z, \phi) \right] \right] + \mathbb{E}_{p(\phi,x,z)} \left[ \log |\det J_f(x)| - \log p(x \,|\, z, \phi) \right],
\end{aligned}
$$

where $J_f(x)$ denotes the Jacobian of $f(\cdot)$ evaluated at $x$, the third line follows from change of measure, and the term $\mathbb{E}_{p(\phi,x,z)} \left[ \log |\det J_f(x)| - \log p(x \,|\, z, \phi) \right]$ is constant with respect to $\theta$. $\qquad\square$

## B. Supplementary Evaluation Metrics

The main performance metric quoted in the main body of the paper is *expected negative log likelihood*, where the expectation is taken over the joint distribution of priors, latents, and observations. This is standard in the literature as it simultaneously measures both accuracy and calibration, it cannot be "gamed" by focusing densities to a singularity around evaluation latents due to the expectation procedure, and it is equal up to an additive constant to the expected KL-divergence with the true posterior (which lower bounds the expected NLL) where the expectation is taken with respect to the joint distribution over priors and observations, with latents marginalised out, as demonstrated by Proposition 3.1. In this way, expected NLL serves as an excellent relative metric with the difference in performance between two methods exactly corresponding to the difference in their expected KL-divergence with the true posterior. Furthermore, other traditional metrics are not typically applicable to amortised settings due to having only a single posterior sample per set of observations. That said, for the Posterior Approximation Studies we choose to provide in Table 4 the supplementary metric of MMD between the true joint distribution over priors and latents, $p(\phi, x) = p(\phi)p(x \,|\, \phi)$, and that implied by the approximate inference method, $q(\phi, x) = p(\phi)\mathbb{E}_{z \sim p(z \,|\, x)} \left[ q(x \,|\, \phi, z) \right]$ computed using a product of exponential kernel embedding over prior and latent samples, to directly measure calibration.

## C. Ablations and Supplementary Analyses

We provide here a series of additional experiments supporting those in the main text.

### C.1. Analytical Scaling Experiment

We present here an exposition of how the method performs in much higher dimensions. To ablate away the scaling performance of GMMs, which can be made arbitrarily good by adding more components, and isolate the performance of the architecture itself, we choose the problem of conditioning on a single 4-dimensional observation with multivariate Gaussian likelihood, and a conjugate 8-component GMM prior thus inducing a closed-form 8-component GMM posterior. Using 8 components in our posterior approximation, the theoretical minimum expected KL-divergence attainable is exactly 0, meaning any discrepancy here is purely induced by an imperfect mapping.

We use an identical model and training setup to the other experiments, and adopt a simple, wide meta-prior here, namely independent standard Gaussian distributions over component means, inverse-gamma (1, 1) distributions over component variances, and a uniform Dirichlet distribution over the component weights. We sweep the dimensionality of the latent variable $x$ from 1 to 24, and compare only against SVI as all other competitors are not capable of modelling multivariate joint posteriors.

Figure 8 shows and explains that while performance inevitably degrades with dimensionality due to accumulation of modelling errors in the greater number of output parameters, DTs remain at least an order of magnitude better than SVI at all dimensionalities.

*Table 4.* Calibration as measured by MMD for Experiments 4.2.1 and 4.2.2
.

| EXPERIMENT | METHOD | MMD |
|---|---|---|
| 4.2.1 PPD | | |
| | PFN | $0.0058 \pm 0.0018$ |
| | TABPFNV2 | $0.0063 \pm 0.0024$ |
| | ACE | $\mathbf{0.0035 \pm 0.0013}$ |
| | DT | $\mathbf{0.0036 \pm 0.0014}$ |
| 4.2.1 HYPERPOSTERIOR | | |
| | SVI | $0.436 \pm 0.013$ |
| | MCMC | $0.116 \pm 0.017$ |
| | PFN | $0.0093 \pm 0.0024$ |
| | TABPFNV2 | $\mathbf{0.0029 \pm 0.0019}$ |
| | ACE | $0.0041 \pm 0.0015$ |
| | DT | $0.0059 \pm 0.0024$ |
| 4.2.2 | | |
| | SVI | $\mathbf{0.0028 \pm 0.0004}$ |
| | PFN | $0.0100 \pm 0.0006$ |
| | TABPFNV2 | $\mathbf{0.0046 \pm 0.0007}$ |
| | ACE | $0.0067 \pm 0.0024$ |
| | DT | $\mathbf{0.0027 \pm 0.0004}$ |

*Table 5.* Both DT-5 and ACE-5 degrade slightly upon testing on the unseen meta-prior, but DT-5 remains accurate and continues to outperform.

| METHOD | TRAINING META-PRIOR EXPECTED KL-DIVERGENCE | UNSEEN META-PRIOR EXPECTED KL-DIVERGENCE |
|---|---|---|
| ACE-5 | $0.0058 \pm 0.0014$ | $0.0132 \pm 0.0010$ |
| DT-5 | $\mathbf{0.0003 \pm 0.0000}$ | $\mathbf{0.0054 \pm 0.0015}$ |

## C.2. Misspecification of Meta-Prior

Generalisation to substantially out-of-support meta-priors is an important open question. To probe this and provide some initial exploratory evidence, we provide an ablation over Experiment 4.1 using a shifted test meta-prior with largely non-overlapping mass. Specifically, we train on the wide meta-prior used in Experiment 4.1, but compute test statistics with respect to the following, out-of-sample meta-prior:

$$\text{rate} \sim \text{InverseGamma}(24, 6)$$
$$\text{scale} \sim \text{InverseGamma}(24, 6)$$

Results for ACE-5 and DT-5 are presented in Table 5.

## C.3. Prior Loss Ablation

One of our more subtle contributions is the prior loss term to the amortised training objective. This loss term has two important effects:

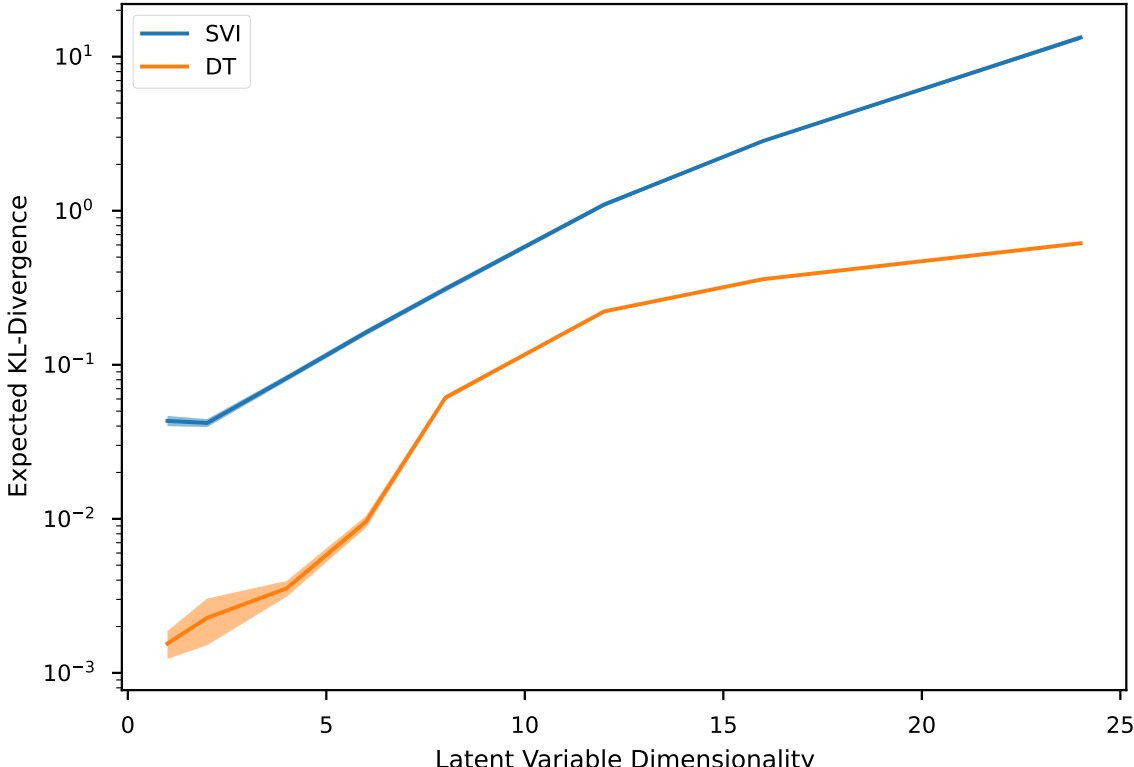

*Figure 8.* Expected KL-divergence against latent variable dimensionality for a simple conjugate experiment. Note that while the expected KL-divergence is not uniformly zero and is increasing in dimensionality for DTs, at all dimensionalities DTs are at least of an order of magnitude better than SVI, indicating that misspecification of the posterior (or indeed choosing a less powerful approximating family) is of considerably greater importance at all dimensionalities. Furthermore, noting the log scale this increase is seemingly polynomial (likely quadratic, as the number of output parameters is quadratic in latent dimension) in dimensionality for DTs, while the increase in SVI and therefore posterior family-misspecification is exponential, as expected.

1. It imbues the model with approximate latent-space conjugacy, a crucial property for recursive application without unembedding from and re-embedding to the latent state.

2. It acts as a regulariser and extra training signal on the primary posterior approximation task.

We demonstrate performance with and without this term on the wide meta-prior variant of Experiment 4.1 in Table 6.

### C.4. Generalisation Over GMM Sizes

One exciting possibility of Distribution Transformers is generalisation at inference-time to an unseen number of GMM components. The architecture is inherently invariant to the number of GMM components used, so we would expect to see some generalisability across the number of GMM components chosen, even if this is fixed during training. We provide here some preliminary evidence of this, in the form of an 8-dimensional linear Gaussian likelihood inference problem, where we train over 10-component GMMs and test over a wider range. The results for this are provided in Table 7.

### C.5. Error Accumulation with Sequence Depth

One inevitability of approximate inference methods used in sequential settings is that over recursive uses of the model, errors will start to accumulate. We thus provide here an ablation based on Experiment 4.3.2 — our most challenging sequential inference experiment. Due to variable scale over sequence depth, NLL alone is not meaningful to measure these effects as it can only compare methods at a given depth rather an across depths. We thus report the *relative excess expected NLL* of DTs against the assumed-ground-truth largest PF used:

*Table 6.* Inclusion of the prior loss term makes a small improvement to the main posterior objective, but more importantly it is critical for ensuring approximate latent-space conjugacy. This conjugacy is crucial for recursive application, as it obviates the need to unembed from one posterior to the next prior.

| PRIOR LOSS | EXPECTED POSTERIOR KL-DIVERGENCE | EXPECTED PRIOR KL-DIVERGENCE |
|:---:|:---:|:---:|
| ✓ | **0.0003±0.0000** | **0.0004±0.0000** |
| ✗ | 0.0004±0.0000 | 0.3020±0.190 |

*Table 7.* Preliminary results indicate that even when trained over a single GMM configuration, DTs may generalise to other GMM component counts *at inference time* without retraining. The DT was trained with 10 GMM components, and experiences only mild degradation upon both halving and doubling the component count.

| NUMBER OF GMM COMPONENTS | EXPECTED POSTERIOR KL-DIVERGENCE |
|:---:|:---:|
| 2 | 0.1494±0.0054 |
| 5 | 0.0985±0.0039 |
| 10* | 0.0873±0.0031 |
| 20 | 0.0946±0.0036 |

$$\text{Relative Excess Expected NLL}_t = \frac{\mathbb{E}_{p(x_t, z_{0:t})}\left[\log q_{\text{PF}}(x_t \mid z_{0:t}) - \log q_{\text{DT}}(x_t \mid z_{0:t})\right]}{\mathbb{E}_{p(x_t, z_{0:t})}\left[\log q_{\text{PF}}(x_t \mid z_{0:t}) - \log p(x_t)\right]}$$

$$= \frac{\mathbb{E}_{p(z_{0:t})}\left[\text{KL}\left[p(x_t \mid z_{0:t}) \,\|\, q_{\text{DT}}(x_t \mid z_{0:t})\right] - \text{KL}\left[p(x_t \mid z_{0:t}) \,\|\, q_{\text{PF}}(x_t \mid z_{0:t})\right]\right]}{\mathbb{E}_{p(z_{0:t})}\left[\text{KL}\left[p(x_t \mid z_{0:t}) \,\|\, p(x_t)\right] - \text{KL}\left[p(x_t \mid z_{0:t}) \,\|\, q_{\text{PF}}(x_t \mid z_{0:t})\right]\right]},$$

where $p(x_t)$ is the marginal density at recursion depth $t$, available in closed form by propagation of $p(x_0)$ through the dynamics. The advantage of such a quantity is that all scale information is normalised by the ratio, and each subtraction converts the NLLs to KL-Divergences by elimination of the differential entropy of the true posterior. This quantity is negative when DTs outperform PFs and vice versa, and assuming DTs and PFs are both perfectly calibrated, simplifies further to $1 - \frac{\text{I}_{\text{DT}}(x_t; z_{0:t})}{\text{I}_{\text{PF}}(x_t; z_{0:t})}$, where $\text{I}_{\text{DT}}$ and $\text{I}_{\text{PF}}$ are the mutual information between state and observations implied by the DT and PF respectively.

Figure 9 demonstrates this relative excess expected NLL against sequence depth for Experiment 4.3.2. This quantity is statistically indistinguishable from 0 until a sequence depth of around 70, then slowly rises thereafter, though remaining small. Even at a sequence depth of 100, DTs retain roughly 95% of the information content of PFs, providing some preliminary evidence that they may be suitable for long-horizon sequential inference problems.

## D. General Architecture Details

Learnable prior embeddings typically consist of a multi-layer perceptron (MLP) with one hidden layer, usually of half the size of the transformer latent space. For GMM priors, the prior embedding acts elementwise and consists of a logarithm applied to the component weight, a Cholesky decomposition then logarithm of diagonal elements applied to the covariance matrix, and a flattening into a vector before passing through the MLP straight to the latent unordered sequence. For general priors, simple transformations are applied to parameters, for example positive parameters are passed through a logarithm, before being passed through the MLP to a single vector in the latent space. Then, distinct MLPs act on this vector to yield the components of the latent unordered sequence.

Learnable observation embeddings always consist only of an MLP, but theoretically could be extended to include inductive biases suitable for the nature of each observation. These embeddings are not processed further, and attend directly to the latent unordered sequence.

A standard transformer decoder setup is used, always consisting of 6 transformer decoder units, typically with a latent space size of 64, an MLP hidden layer size of 2048, and 8 attention heads. Layer norm was also chosen for its numerous benefits (Ziomek et al., 2025).

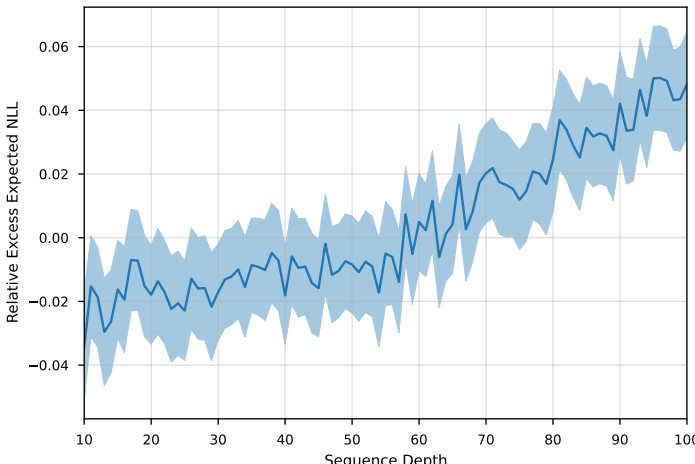

*Figure 9.* Relative Excess Expected NLL of DTs against PFs on Experiment 4.3.2. The accumulated error remains small even at moderate sequence depths, and is statistically indistinguishable from 0 up until a depth of around 70.

The unembedding is almost an exact reverse to the GMM prior embedding, acting elementwise first with a learnable MLP, then the inverse of the Cholesky-log-flatten transform described earlier for the component covariance matrix. A cross-component softmax is then applied to the unembedded weight logits to yield the GMM.

Note that for numerical stability, GMMs were always parametrised directly by the Cholesky decomposition of the covariance matrix.

For PFNs, an identical setup was used wherever possible; Observations were embedded with the same embedding architecture, and the transformer encoder used identical hyperparameters to the DT for each experiment.

For ACE, we also tried to use as similar a setup as possible; for this reason we used the GMM heads and used the same number of components and sample space transformations as in DTs. Additionally, since we are only interested in inference of certain variables in the problem, we did not randomise the variable with respect to which NLL loss is computed and instead kept it fixed to the variable of interest. We provide prior hyperparameters as additional latent variables on each problem.

For TabPFNv2, we simply used the pretrained foundational model provided by the authors and conditioned it on 10,000 samples (which is the maximum recommended by the authors). We provided prior hyperparameters as additional observations the model was conditioned on.

## E. General Training Details

All experiments were trained using the Adam optimiser without weight decay or dropout (overfitting here is impossible as the model sees each sample only once). A cosine-annealing-with-warmup (5 epochs) learning rate scheduler was used. This training scheme was adopted for both DTs and PFNs.

SVI also used the Adam optimiser without weight decay or dropout, directly optimising variational parameters via backpropagation through ELBO using Adam. An exponential learning rate scheduler was used.

Experiments 4.1, 4.2.2, and 4.3.1 were all carried out on an NVIDIA RTX 2080 Super laptop GPU (8GB VRAM), while Experiments 4.2.1 and 4.3.2 were carried out on an NVIDIA RTX 3090 GPU.

## F. Details for Analytical Verification Study (Experiment 4.1)

Here, for both the inverse-gamma prior's rate and scale parameters, we adopt an $\text{InverseGamma}(4, 6)$ meta-prior and an $\text{InverseGamma}(10000, 20000)$ for the wide and narrow meta-priors respectively. Both of these meta-priors have the same mean, but the narrow meta-prior is effectively singular.

*Table 8.* Training hyperparameters and statistics for all experiments. An epoch always consists of 100 batches, except for in the case of Experiment 4.3.2 where memory constraints required smaller batches with 500 batches per epoch. Hyperparameters were obtained by limited manual tuning, adjusted until training was stable in all cases (although note that none of the methods hear appear to be particularly sensitive to hyperparameter choice, generically requiring only a sufficiently small learning rate and training until apparent convergence). For SVI, the total samples reported account for batching, so the number of samples used by each SVI problem will be a factor of the number of test problems lesser. Parameters are listed vertically, corresponding to DTs (top), PFNs (second from top), SVI (third from top) and ACE (bottom). For Experiment 4.1, only DT-5 is reported but training hyperparameters are identical for DT-2. For experiment 4.3.1 only DT is reported. For experiment 4.2.1, two PFNs are trained (one for each dimension, as PFNs do not trivially support multivariate distributions), so two figures are quoted when necessary (PPD + hyperposterior). Model sizes are not provided for VI as this is negligible.

| EXPERIMENT | LR | BATCH SIZE | EPOCHS | TOTAL SAMPLES (M) | TRAINING TIME (S) | MODEL SIZE ($10^6$ PARAMS) |
|---|---|---|---|---|---|---|
| 4.1 | 0.005 | 5000 | 20 | 10 | 188 | 0.43 |
| | 0.005 | 5000 | 20 | 10 | 247 | 0.29 |
| | 0.1 | 10 | 100 | 10 | 137 | — |
| | 0.001 | 5000 | 20 | 10 | 171 | 0.98 |
| 4.2.1 | 0.0001 | 2000 | 200 | 40 | 4300 | 18.8 |
| | 0.0001 | 1000 | 200+100 | 20+10 | 730+369 | 13.9 |
| | 0.03 | 10 | 1000 | 100 | 131 | — |
| | 0.0001 | 5000 | 40 | 80 | 4160 | 23.9 |
| 4.2.2 | 0.001 | 5000 | 20 | 10 | 222 | 1.80 |
| | 0.0001 | 4000 | 25 | 10 | 290 | 1.63 |
| | 0.01 | 10 | 100 | 10 | 1036 | — |
| | 0.0001 | 5000 | 20 | 10 | 356 | 2.31 |
| 4.3.1 | 0.001 | 5000 | 150 | 75 | 3720 | 1.72 |
| 4.3.2 | 0.001 | 1000 | 130 | 65 | 7564 | 1.72 |

An analytical expression for the ground-truth posterior distribution is obtainable via conjugacy. Specifically, given a measurement $\sigma^2$ and prior parameters $\alpha_0$ and $\beta_0$, the parameters of the also inverse-gamma posterior are given by:

$$\alpha = \alpha_0 + \frac{1}{2}, \text{ and}$$
$$\beta = \beta_0 + \frac{(z - \mu)^2}{2},$$

where $z$ is the observation value and $\mu$ is the known observation mean, which we arbitrarily set to 0.

## G. Details for Gaussian Process Joint PPD and Hyperposterior Study (Experiment 4.2.1)

For the meta-prior in this experiment, we assign a narrow, uniform prior over the PPD's prior such that it is marginally everywhere a standard normal. We assign an inverse-gamma prior over lengthscale, whose hyperparameters are sampled from a uniform meta-prior, which is over $[100, 105]$ for concentration and $[100, 700]$ for rate.

Note that for this experiment, the PPD prior also, in conjunction with the lengthscale $l$, defines the hyperparameters of the GP's prior mean function and RBF covariance function used here. Specifically, the GP's mean function is set to a constant at the PPD prior's mean, and the output scale of the RBF covariance function is set to the standard deviation of the PPD's prior.

The sampling procedure for this experiment, using 5 observations (each observation being 5-dim input and 1-dim function value), was as follows:

1. Sample $\phi$ from meta-prior

2. Sample $y \sim \mathcal{N}(\phi_\mu, \phi_{\sigma^2})$

3. Sample $l \sim \text{InverseGamma}(\phi_\alpha, \phi_\beta)$

4. Sample $x \sim \text{Uniform}(0,5)^5$

5. Sample $X \sim \text{Uniform}(0,5)^{5 \times 5}$

6. Sample $Y \sim \mathcal{N}(\phi_\mu, k(X, X; l, \phi_{\sigma^2}))$

7. Construct $z$ as concatenation of $X$ and $Y$ in the last dimension, along with the query point $x$.

Two observation embeddings are defined, one acting on each $X$-$Y$ element pair, and one acting on the query point $x$. Each have one hidden layer of size 128.

The prior embedding consists of two MLPs, acting on the PPD prior and the lengthscale prior respectively, each with one hidden layer of size 128, with 32 outputs. These outputs are concatenated into a single vector of size 64, before being passed through another set of distinct, parallel MLPs as before to yield the required length-10 latent unordered sequence.

## H. Details for Quantum System Parameter Inference Study (Experiment 4.2.2)

This experiment is a probabilistic adaptation of a well-known two-level quantum system problem (Nielsen & Chuang, 2010; Schorling et al., 2025). We adopt a beta prior, with an InverseGamma$(4, 6)$ meta-prior over each parameter.

Observations are sampled using a GPU-implemented simulation, which accepts a nominal initial state $|\hat{\psi}_0\rangle$ and nominal measurement time $\hat{t}$. We model uncertainty in the initial state preparation by, treating $|\hat{\psi}_0\rangle$ as a 2-element vector, sampling an actual initial state $|\psi_0\rangle \sim \mathcal{N}(|\hat{\psi}_0\rangle, 0.01I)$, and renormalising such that $\langle\psi_0||\psi_0\rangle = 1$. We also model uncertainty in measurement time by sampling $t \sim \mathcal{N}(\hat{t}, 0.0025)$. We then solve Schrödinger's equation to give $|\psi_t\rangle$, and take a measurement by sampling from the Bernoulli distribution associated with $|\psi_t\rangle$. For SVI, the likelihood of this observation is estimated stochastically with 10 samples of the initial state and measurement time.

A set of varying nominal initial states and measurement times is fixed, and with each problem sampling an observation from each is obtained. Each pair of initial state and measurement time is provided its own learnable embedding, all of standard construction. The prior embedding and posterior unembedding is also standard.

## I. Details for Bayesian Sensor Fusion Study (Experiment 4.3.1)

For this experiment, we use the following meta-prior over a 4-component GMM:

$$
\begin{aligned}
w &\sim \text{Dirichlet}(0.25 \cdot \mathbf{1_4}) \\
x_i &\sim \mathcal{N}(\mathbf{0_4}, 4I_{4\times 4}) & \forall i = 1 \ldots 4 \\
\Sigma_i &\sim \text{Wishart}(5, 4I_{4\times 4}) & \forall i = 1 \ldots 4,
\end{aligned}
$$

where $\mathbf{1_4}$ and $\mathbf{0_4}$ refer to a 4-element vector consisting of only ones and zeros respectively.

We use two realistic observation models: A rangefinder subject to maximum range constraints, range-dependent Gaussian noise centred about the true range, exponentially distributed early measurements from unexpected objects, and uniformly distributed readings due to sensor failure, and bearing measurement subject to Gaussian noise and cyclic discontinuity (ie a wrapped normal distribution). Figure 10 demonstrates the complexity of the rangefinder observation model, and the parameters used in this experiment are presented in the caption.

We report iteration time per 100 series. Concretely, this refers to the prediction, update, and density fitting, of a single time step batched across 100 series.

We use a motion model representing damped velocity subject to independent, normally distributed accelerations in each direction. Formally, we adopt the following discrete-time dynamical system:

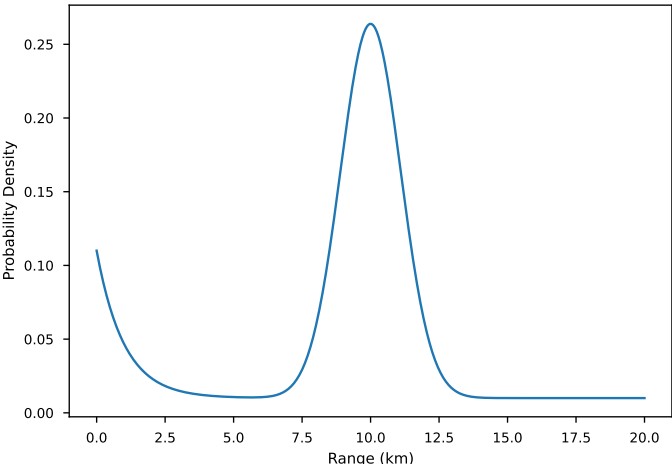

*Figure 10.* Example likelihood for rangefinder observation model, conditioned on a range of 10km. We assume a maximum range of 20km, a true observation standard deviation of $0.1(\text{range} + 1)$km with marginal probability 0.7, an early collision decay rate of $1\text{km}^{-1}$ with marginal probability 0.2, and a uniform sensor failure on $\text{Uniform}(0, 20\text{km})$ with marginal probability 0.1. While this model is realistic, the uncertainties involved are amplified compared to those seen in reality. This was done to further illustrate the capabilities of our model.

$$x_{t+1} = \begin{bmatrix} 1 & 0.63 & 0 & 0 \\ 0 & 0.37 & 0 & 0 \\ 0 & 0 & 1 & 0.63 \\ 0 & 0 & 0 & 0.37 \end{bmatrix} x_t + \begin{bmatrix} 0.1 & 0.0 \\ 0.2 & 0.0 \\ 0.0 & 0.1 \\ 0.0 & 0.2 \end{bmatrix} w_t$$

$$w_t \sim \mathcal{N}(\mathbf{0_2}, I_{2\times 2})$$

$$x_0 \sim \mathcal{N}(\mathbf{1_4}, I_{4\times 4})$$

## J. Details for Factor-Structure Stochastic Volatility Study (Experiment 4.3.2)

For this experiment, we use the same meta-prior as in Experiment 4.3.1, with the latent variable being the log-volatilities of the 10 factors.

Observations are sampled from the following likelihood:

$$p(z \mid x) = \mathcal{N}\left(\mathbf{0_{30}}, F \operatorname{diag}(\exp(x)) F^{\top} + \sigma_{\text{idio}}^2 I_{30\times 30}\right),$$

where $F \in \mathbb{R}^{30\times 10}$ is a known factor loading matrix with elements sampled from the standard normal, $\sigma_{\text{idio}}^2 = 1$ is the variance of the assumed-isotropic idiosyncratic component, $\exp$ is applied element-wise, and diag embeds a vector as a diagonal matrix.

Log-volatilies $x$ evolve according to the following discretised Ornstein-Uhlenbeck process:

$$x_{t+1} = \mu + \alpha \odot (x_t - \mu) + \epsilon$$

$$\mu \in \mathbb{R}^{10} = \begin{bmatrix} 0.99 & 0.98 & 0.97 & 0.96 & 0.95 & 0.94 & 0.93 & 0.92 & 0.91 & 0.9 \end{bmatrix}^{\top}$$

$$\alpha \in \mathbb{R}^{10} = \begin{bmatrix} -1.0 & -0.8 & -0.6 & -0.4 & -0.2 & 0.2 & 0.4 & 0.6 & 0.8 & 1.0 \end{bmatrix}^{\top}$$

$$\epsilon \in \mathbb{R}^{10} \sim \mathcal{N}\left(\mathbf{0_{10}}, \operatorname{diag}(\sigma^2)\right)$$

$$\sigma \in \mathbb{R}^{10} = \begin{bmatrix} 0.1 & 0.2 & 0.3 & 0.4 & 0.5 & 0.6 & 0.7 & 0.8 & 0.9 & 1.0 \end{bmatrix}^{\top},$$

where $\odot$ denotes the element-wise product.

The data for Figure 7 is obtained by running PFs with a particle count varying from 500 to 500,000. Due to the extreme memory requirements of the larger PFs, we substitute the Gaussian KDE here for a moment-matched multivariate Gaussian. Furthermore, the memory requirements prohibited processing series in parallel. Thus, for fairness the processing frequencies quoted here are derived from the time taken for each method to process a *single* series, rather than a batch of series. Note that these issues *do not affect DTs*, who are still more than capable of large batch processing at speed without memory issues. Expected NLL values are still computed over a batch of series, as in Experiment 4.3.1.

