# OpenReview forum: "Distribution Transformers: Fast Approximate Bayesian Inference With On-The-Fly Prior Adaptation"
_ICML.cc/2026/Conference — ICML 2026 spotlight_

### Official Review · Reviewer_7RzB · 2026-03-06

**Soundness:** 3
**Presentation:** 2
**Significance:** 2
**Originality:** 3
**Overall Recommendation:** 4
**Confidence:** 3

**Summary:**

The paper proposed Distribution Transformers (DTs), a transformer-based architecture for (prior) amortized Bayesian inference that jointly represent priors and posteriors as Gaussian Mixture Models. Observation are encoded via an additional embedding network into a set of tokens to which an transformer decoder attends to via cross-attention. Priors are represented as a set of learnable embeddings per GMM component, these are pushed through the decoder yielding a set of posterior embeddings. A learnable unembedding network (shared between prior and decoder) learns to (re)construct GMM components from the tokens, yielding a GMM approximation of the posterior. As both prior and posterior remain GMM this yields an "approximately conjugate" inference. The paper then demonstrate that DTs can do (prior) amortized inference in simple  tractable 1d toy examples (Gamma-Gaussian + GP) and a quantum parameter inference problem showing improved performance compared to baselines (e.g. Prior-Fitted Networks PFNs).

**Compliance With Llm Reviewing Policy:**

Affirmed.

**Final Justification:**

The major concerns have been adressed or the identified limitations have been promised to be added to the manuscript.

**Key Questions For Authors:**

- Maybe to clarify the difference to ACE: it uses a autoregressive decoder + 1d GMM for conditionals from tokens. In contrast  DTs use a full attention decoder + decoding multivariate Gaussians + logit from each token?
- Do DTs generalize over the number of components i.e. principally the transformer allows to put in any number of components for the prior, no?
- Is e.g. DT-2 / DT-5 in the paper the same trained network, or is it separately trained?

**Limitations:**

There is no limitation section, some point that should be discussed:
- Scalability is unclear. The method seems best suited to low- or moderate-dimensional latent spaces, since both full-covariance GMM decoding and attention over mixture components may become expensive as dimension or component count grows. There should be an discussion on when the method is expected to work well, and when it might struggels.
- Dependence on the training meta-prior is under-discussed. Good performance may require test-time priors and recursively generated posteriors to stay within the support of the meta-prior seen during training.
- No analysis of recursive error accumulation. This is a central limitation given the paper’s sequential-inference motivation: repeated posterior-to-prior reuse could lead to drift over long horizons.

**Strengths And Weaknesses:**

## Strength
- The paper addresses an important and underexplored problem in amortized Bayesian inference, namely **efficient prior amortization** without retraining. The idea of maintaining both prior and posterior in the same approximate parametric family is interesting and, to my knowledge, novel in this context.
- The manuscript is well-written and the methodolgy is sound.
- The motivation for **sequential inference** is compelling: if the posterior can be reused directly as the next prior, this gives DTs a practical advantage over approaches that only output task-specific posterior summaries. The paper is generally well written, and the high-level architecture is easy to follow.


## Weaknesses
- The presented empirical evidence is somewhat weak.
	-  Figure 3: Shows a "prior-aware" DT approximation vs. a "prior-unaware" PFN implementation. And *expectedly* the PFN fails, because given its loss it will learn the *marginalized* posterior. So what is the point of this comparison?
	- The supplement indicates, TabPFNv2 and ACE is conditioned on hyperparameters so why not also PFN to make this a fair comparison (although I would recommend to replace this anyway, see second point).
- There is **no analysis of error accumulation** under long recursive rollouts, which is especially important for sequential inference.
	- Any conjugate family (as e.g. the one in 3) would be suitable to investigate this e.g. even in the IID setting p(x| z_1, z_2) ~ p(z_2| x)p(x| z_1) (i.e. to get the posterior of z_1 and z_2 we can just sequentially apply the DT in Fig.3 (reusing the current posterior as new prior).
	- This also requires a **well chosen metaprior** i.e. any posterior needs to be within training metaprior. This should limitation should be discussed in the manuscript.
- Scalability + complexity remains unclear and is not discussed/ablated. Limitations are not really discussed.
	- Experiments are limited to *lower dimensional* settings.
	- ACE, to my knowledge, scales cubic (quadratic via kv caching) in dimension and constant in the number of mixture components. From each token it has to decode O(K) values.
	- DTs  scale quadratic in the number of mixture components + constant/quadratic in dimension (depending on covariance parameterization, or?). From each token DTs also have to decode O(d^2) values (for full covariance)
	- This has several implication i.e. DTs need high state-dimension to be applicable in high dimension, no? (because one has to decode many parameters from one token)

---

> ### Author Rebuttal · Authors · 2026-03-27
>
> We thank the reviewer for the careful and constructive review, and for recognizing the importance of the problem, the novelty of the approach, and the motivation for sequential inference. We agree that the empirical framing and limitations discussion should be clearer.
>
> # Empirical scope
> Our experiments serve three distinct roles: (i) controlled illustrations of prior adaptation (Figs. 3---4), (ii) competitive benchmarks on tractable/intractable problems (Tables 1---2), and (iii) sequential inference settings (Sec. 4.3), where approximate conjugacy is required for recursive reuse. We will revise the paper to make this structure explicit.
>
> # Fig. 3 / PFN fairness
> We agree Fig. 3 should not be read as the main benchmark. Its purpose is to illustrate a) the expected failure mode of a prior-unaware amortizer under a wide meta-prior: it learns the marginalized posterior, as the reviewer points out, and b) the output-efficiency of GMMs relative to Riemann distributions in modelling smooth posteriors. The substantive comparisons are in Tables 1---2, where we compare against PFN/TabPFNv2, ACE, SVI, and MCMC where appropriate. As addressed in the paper, a rudimentary form of prior conditioning for PFN-like models can be added via extra inputs, but this does not address the key issue here: preserving a reusable posterior in the same family as the prior.
>
> # Recursive error accumulation
> We agree this is important. While Sec. 4.3 already applies DT recursively over long horizons, we ran an additional ablation for Exp. 4.3.1 measuring DT excess expected NLL over a PF baseline normalized by excess expected NLL of the marginal density over the same baseline versus recursion depth:
>
> Recursion Depth|Relative Excess Expected NLL
> |-|-|
> 25|$-0.023\pm 0.016$
> 50|$-0.008\pm 0.015$
> 75|$0.012\pm 0.016$
> 100|$0.048\pm 0.017$
>
> While any approximate method will eventually accumulate errors, our results suggest that the effects of error accumulation may be negligible up to even moderate depths.
>
> To remove any ambiguity from approximate baselines, we will additionally include a long-horizon linear-Gaussian experiment where the Kalman filter provides the exact posterior, allowing us to measure true error accumulation analytically.
>
> # Meta-prior dependence
> We fully agree this should be stated more explicitly. DT is designed to generalize across the family of priors represented by the training meta-prior; substantial out-of-support prior/posterior shift remains an underexplored area at present. We refer the reviewer to our response to Reviewer 3sGq, where we provide an additional ablation on shifted meta-priors.
>
> # Scalability / dimensionality
> We agree DT is currently best suited to low-to-moderate dimensional latent spaces. The main costs are attention over $k$ mixture components and decoding multivariate Gaussian parameters. We will clarify this limitation. At the same time, Appendix C already includes a latent-dimension sweep from 1 to 24 while controlling away GMM approximation error; Exp. 4.3.2 also studies a 10D stochastic-volatility model with 30 observations per timestep.
>
> # Difference to ACE
> We will clarify this more explicitly: ACE uses an autoregressive factorization into 1D conditionals, whereas DT processes GMM components jointly via full attention and decodes multivariate Gaussian components plus mixture weights. This is what allows DT to maintain an approximately conjugate multivariate posterior family for sequential reuse.
>
> # Generalization over the number of components
> In the current paper, DT-2 and DT-5 are separately trained models, and we will clarify this. However, we also ran a small 8D ablation suggesting a trained DT (in this case trained with 10 components) can be used with a different number of components at inference time:
>
> $k$ | Expected KL-Divergence
> |-|-|
> 2 | $0.1494\pm 0.0054$
> 5 | $0.0985\pm 0.0039$
> 10*| $0.0873 \pm 0.0031$
> 20 | $0.0946\pm 0.0036$
>
> These results suggest that, although the current implementation is trained with a fixed $k$, the learned representation is not tightly tied to that exact cardinality: performance is best near the training value and degrades only modestly when evaluated with fewer or more components in this controlled setting, indicating promising but still preliminary generalization across $k$.
>
> # Limitations
> We agree a dedicated limitations paragraph is needed. We will add one covering dependence on the training meta-prior, possible long-horizon drift, computational scaling with $k$ and latent dimension, and the fact that the current implementation is most natural for low-to-moderate dimensional latent spaces.
>
> We thank the reviewer again for the helpful feedback. Taken together, the new ablations directly address the main empirical concerns in this review: meta-prior robustness, recursive stability, and representation flexibility; We believe the remaining issues are therefore primarily about clearer framing and limitations, which we will address in revision.

---

> > ### Author Rebuttal · Reviewer_7RzB · 2026-04-02
> >
> > I thank the authors for their response. The additional experiments have resolved my major concerns. The across-K generalization already looks quite promising and could likely be improved further through mixed sequence-length training or related alternatives.
> >
> > More specifically, for higher-dimensional posteriors, or posteriors with unusual geometries, a mixture of multivariate Gaussians can quickly reach its expressive limit compared with an AR approach. The above direction may help address this issue.
> >
> > The authors also acknowledge the limitation regarding scaling to large latent dimensions and have indicated that they will add this discussion to the manuscript. Since there are several applications and use cases where, in my opinion, the proposed approach seems quite effective, I will increase my score.

---

### Official Review · Reviewer_Shcw · 2026-03-07

**Soundness:** 3
**Presentation:** 4
**Significance:** 4
**Originality:** 3
**Overall Recommendation:** 6
**Confidence:** 5

**Summary:**

The authors devise a transformer architecture that maps from a Gaussian mixture model and a set of observations to another Gaussian mixture model. Self attention is performed between mixture component parameters, and cross attention is performed between the mixture components and the observations. In a similar spirit to PFNs, the model is meta-trained across a collection of synthetic datasets coming from some user-specified data-generating process. Once trained, the framework enables users to condition arbitrary prior distributions (in the form of a GMM) on data to obtain posteriors. The approach is elegant, fast, and highly performant. It represents a significant and exciting contribution to the field.

**Compliance With Llm Reviewing Policy:**

Affirmed.

**Final Justification:**

The only concerns I had with the paper were of an editorial nature, and the authors have addressed all of these in their rebuttal. Having reviewed the paper twice now, I am absolutely certain in my view that it is an excellent paper and I would strongly recommend it to be considered for Spotlight/Oral. I believe the contribution is significant to the whole probabilistic machine learning community, and that it has the potential for lots of interesting research to stem from it.

**Key Questions For Authors:**

Q1.) What is the difference between a meta-prior and a hyper-prior? Does a hyper-prior induce a meta-prior? Can we say that a hyper-prior is a way to *parameterise* a meta-prior?

**Limitations:**

Limitations discussed, yes.

**Strengths And Weaknesses:**

### Strengths:

- The writing quality and overall clarity of the narrative is excellent.

- For practitioners of Bayesian inference in arbitrary domains, the contributions of this paper are potentially very significant.

- The claims are clearly met by the comprehensive experimental validation in both synthetic and real-world environments.

- Sufficient experimental details are provided for readers to reproduce the experiments.

- The theory and connection to related work are sound.

- This is subjective of course, but for me the paper is plainly just very exciting.

### Weaknesses / Suggestions:

I was a reviewer for this paper in a previous submission. I am pleased to see that almost all of my concerns and suggested changes have been addressed. For example, one of my more major criticisms of the paper in the previous round of peer review was that the paper offered little in the way of exploring DTs' effectiveness in higher dimensional settings. However, the newly added Appendix C and Figure 8 now provide exactly that investigation, which I imagine many readers will value. Since I have no serious concerns left, here I provide a series of minor ways in which I think the manuscript could be improved.

a.) The way in which the *Inputs* are listed in Algorithm 1 could be improved for better clarity. My recommendation would be "A joint distribution $p(\phi, x, z)$ over priors, latent variables, and observations; the number of training iterations $m$; the batch size $b$; the number of GMM components $k$; and a mapping $f$ form the sample space of interest to the sample space of the GMM.".

b.) I think the phrase "prior amortization" is nonsensical. When we refer to amortised inference, we mean that the computational cost of performing approximate probabilistic inference is *spread-out* (amortised) between the current task of interest and all the tasks seen during training. A prior is just a probability distribution and there is no inherent computational cost associated with it, so it doesn't make sense to *amortise a prior*. DTs amortise inference across tasks *as well as priors*, so that is the language that I would use. It looks like various authors have already started to use the term "prior amortization", so I would understand if the present authors decide to keep using the term despite its flaws.

c.) The reference to [the PFN paper](https://openreview.net/forum?id=KSugKcbNf9) incorrectly spells the first author's surname as "Muller". This should be corrected to "Müller".

d.) In line 5 of section 2.1, "At its core" should be changed to "At their core".

e.) In section 2.2, "with methods such as [...] variational methods" would read slightly better as "with methods such as [...] variational approaches".

f.) In the final paragraph on page 4, the citation to the PFN paper should be within a `\citet{...}` environment rather than a `\citep{...}`.

g.) In proposition 3.1, "Proof in Appendix A" should not be part of the proposition.

h.) The penultimate paragraph of section 4.1 could do with some cleaning up. "while much bigger of a model" could be something like "while much bigger". "those based on Riemannian histogram" reads quite poorly, maybe something like "those with Riemannian predictives" could be better (i.e., it is the *distribution* that is Riemannian rather than the histogram itself, and also the Riemannian object in question needs to be in the plural).

i.) American english is used throughout the paper, with words such as optimise, maximise, or amortise being spelt with a "z" instead of an "s". If American english is to be used, it should be used everywhere. But in the first paragraph of section 4.2.1, the authors use the British spelling of "modelling". This should therefore be changed to "modeling" (as much as it pains me to say so...). The same applies to other uses of "modelling" in the paper.

j.) In the second paragraph of section 4.3, the claim that EKF and PF represent the industry standard should be supported with a suitable citation.

k.) In the second paragraph of section 4.3, "as- sumptions" should be "assumptions".

l.) In the third paragraph of section 4.3, "Firstly" should not have a capital first letter.

Once again, I congratulate the authors on their excellent work and I look forward to discussing anything further with them should they wish to.

---

> ### Author Rebuttal · Authors · 2026-03-27
>
> We thank the reviewer for the very positive and thoughtful review, and especially for noting the improvements relative to the previous version. We are delighted that the paper’s clarity, practical relevance, and higher-dimensional evaluations came across clearly.
>
> # Meta-prior vs. hyper-prior #
> We agree this distinction should be clarified. In our terminology, a hyper-prior is part of a single Bayesian model hierarchy, i.e. a prior over parameters of a prior within one task, and is thus itself part of the prior family we wish to amortize over. A meta-prior, by contrast, is the distribution over task-specific priors used to generate the family of inference problems seen during amortized training. Mathematically, a hyper-prior can induce such a distribution if it is used to sample priors across tasks, so there is clear overlap; the distinction we intend is primarily functional: in this paper, the meta-prior defines the training distribution over priors that DT should generalize across. We will revise the text to make this explicit.
>
> # Terminology #
> We appreciate the reviewer’s point about ‘prior amortisation’. Our intended meaning is that inference is amortized over a family of priors, rather than that the prior itself is amortized. To improve clarity, we will consider revising the wording toward phrases such as ‘prior-flexible amortized inference’, while keeping consistency with related literature.
>
> # Editorial suggestions #
> We thank the reviewer again for the careful reading. We will incorporate all of the listed edits, including the Algorithm 1 wording, citation/style fixes, spelling consistency, and the supporting citation for EKF/PF in Section 4.3.

---

> > ### Author Rebuttal · Reviewer_Shcw · 2026-04-01
> >
> > Thanks for your response and for clarifying the difference between a meta- and hyper-prior for me, I very much appreciate that.
> >
> > I hope to see your paper accepted!

---

### Official Review · Reviewer_RRZ3 · 2026-03-11

**Soundness:** 3
**Presentation:** 4
**Significance:** 4
**Originality:** 3
**Overall Recommendation:** 5
**Confidence:** 4

**Summary:**

The paper introduces "Distribution Transformers" (DTs) as a novel architecture for amortized Bayesian inference. Traditional amortized methods typically assume a fixed prior during training, entailing costly retraining or fine-tuning if the prior changes at test time. They may not also preserve the distributional family structure between the prior and posterior, limiting their use in sequential filtering applications.

The DT architecture overcomes these shortcomings by representing both prior and posterior distributions as Gaussian Mixture Models (GMMs), where the GMM components are treated as tokenised, unordered sequences. The transformer applies self-attention within the GMMs components, and cross-attention to the observed data, mapping the prior to a posterior within the same family.

The approach is evaluated across several test problems of varying nature where DTs meet or exceed standard inference methods (SBI and prior-fitted networks) while offering substantial improvements to inference time.

The paper is well-written and presented, and as a reader I appreciated the exposition and the elegant GMM-based formulation. As a practitioner in Bayesian inference, the line of research is highly relevant to the community, and applicable to several test problems across disciplines.

**Compliance With Llm Reviewing Policy:**

Affirmed.

**Final Justification:**

I appreciate the authors engagement during the rebuttal - the proposed changes further strengthen the paper and reinforce my initial assessment of an 'Accept'. I see there is general consensus among reviewers towards acceptance, which I support as well.

**Key Questions For Authors:**

1. Architectural Modularity and Latent Bottlenecks

The use of separate learnable embeddings for observations and prior parameters is a clear strength, as it allows for customized modules to capture specific inductive biases. However, to facilitate cross-attention, these disparate inputs must eventually be mapped to the same latent dimension.

- In cases where observation data is high-dimensional (e.g., images or video) while prior parameters are low-dimensional, does this forced dimensionality matching create a significant bottleneck or limit the expressiveness of the more complex input?
- Have the authors experimented with varying the transformer latent dimension $d$ (currently $d=64$) to see if it significantly impacts the "resolution" of the GMM components when conditioned on complex data?

---

2. Meta-Prior Assumptions and Data Efficiency

The proposed framework relies on access to the joint distribution $p(\phi, x, z)$, requiring a well-defined meta-prior $p(\phi)$, a prior $p(x|\phi)$, and a likelihood $p(z|x)$.

- How robust is the proposed method to misspecification of these distributions during training, and how would it handle real-world applications where the exact joint distribution is unknown (with no funtional access) or even misrepresented in the training dataset?
- Given that transformers are typically data-intensive, it could be interesting for the reader to see ablation studies or insights regarding the data efficiency of the training process, specifically the number of joint samples required for the model to generalize across the meta-prior space.

---

3. Characterising Sources of Approximation Error

While the method is end-to-end, errors can find entry at two stages: the GMM's ability to approximate the prior family and the transformer's accuracy in mapping those components to the posterior. Could the authors comment/provide an ablation to identify which part is the dominant error contributor? For example, comparing the KL-divergence of the learned mapping against the theoretical minimum error for a GMM with $k$ components would be highly informative. This is not a must-do but would strengthen the paper even further.

---

4. Computational Scaling and Hyperparameter Efficiency

When facing more complex "hard" problems, what is the efficiency of trading performance for computation? Specifically, how do the number of GMM components $k$ and the number of transformer parameters scale in terms of performance boost versus inference time?  This might be directly relevant to the disentanglement of the sources of the approximation error.

---

5. Role and Necessity of the Prior Loss Term

The combined objective $l_{\theta}^{\prime} = l_{\theta}^\text{prior} + l_{\theta}$ is used to ensure consistency in the shared latent space. Is optimizing for the posterior alone ($l_{\theta}$) insufficient? Since the GMM unembedding is shared between the prior and posterior, wouldn't the prior representation naturally be well-trained through the posterior updates?

**Limitations:**

Yes

**Strengths And Weaknesses:**

**Soundness**: The paper stands on strong theoretical bases for the training objective, with the proposed loss function shown to be equivalent to the expected KL-divergence between the GMM approximation and the true posterior. Permutation equivariance is also preserved by removing positional encodings.

**Presentation**: is excellent in my view - clearly written with good structure and easy to follow as a reader. The motivation, approach and results are clear and with good readability.

**Significance**: is high as the paper approaches important, longstanding bottlenecks in probabilistic ML, and potentially can have a large impact in several downstream applications - particularly in robust decision making for science/engineering where the prior may need updating based on iterative experiments/workflows.

**Originality**: I would rate the proposed DT architecture high on originality/novelty due to its prior-flexibility, distribution tokenisation (treating an entire distribution as an unordered sequence of tokens/GMM components), and being able to map to posteriors within the same GMM family.

---

> ### Author Rebuttal · Authors · 2026-03-27
>
> We thank the reviewer for the detailed and thoughtful review, and for the strong assessment of the paper’s soundness, clarity, significance, and originality. We are especially glad that the GMM-based formulation and the practical relevance resonated.
>
> # Architectural modularity / possible latent bottleneck #
> We agree this is an important design question. In our current experiments, observations are either low-dimensional or are represented as sets of low-dimensional elements that are embedded token-wise, so we did not observe an empirical bottleneck from the shared latent width. More generally, the prior and observation embeddings are modular and may have very different input dimensionalities and architectures; they only need to emit tokens in a common latent space for attention to operate. For genuinely high-dimensional modalities such as images or video, the intended extension would be to replace the simple MLP observation embedding with a modality-appropriate encoder before projection into the DT latent space, which accordingly should have sufficient size to match the intrinsic dimensionality of the data, as in standard representation learning. We will clarify this point in the paper.
>
> # Effect of transformer latent dimension #
> In the current submission we use a standard decoder configuration that typically has latent width 64, 6 decoder blocks, 8 heads, and MLP width 2048. We have not yet included a systematic ablation over latent width, so we do not want to overclaim here. Our expectation is that, as in standard representation learning, larger latent width should help when the conditioning signal or posterior geometry is more complex, at increased computation cost. We will add this as a limitation.
>
> # Meta-prior assumptions / misspecification #
> We will clarify that DT amortizes inference for a specified hierarchical Bayesian model $p(\phi)p(x|\phi)p(z|x)$. If the prior, likelihood, or meta-prior are misspecified, DT will efficiently approximate inference for that misspecified model; it does not itself resolve model misspecification. Regarding shift in the meta-prior family, we address this in our response to reviewer 3sGq with an additional small ablation showing some robustness to moderate meta-prior shift, while treating substantial out-of-support generalization as an open problem.
>
> # Data efficiency #
> We agree this would be valuable to characterize better. The current paper, as in much amortized Bayesian inference work,  assumes efficient sample access to the joint distribution and focuses on test-time inference efficiency rather than sample-efficiency of training; indeed, we visit training sample only once, rather than efficiently reusing samples. This is especially relevant in settings where likelihood sampling requires expensive simulation. We will clarify this limitation and connect it to emerging work on simulation-efficient inference.
>
> # Sources of approximation error #
> Appendix C already partially disentangles these sources. There we construct a conjugate setting in which, with the chosen number of GMM components, the theoretical minimum approximation error from the GMM family is exactly zero; any remaining error is therefore due purely to the learned mapping. Separately, Figure 1 and Table 1 illustrate how increasing the number of GMM components improves approximation quality, showing the role of posterior-family expressiveness. We will make this connection more explicit in the revision.
>
> # Scaling with number of GMM components / model size #
> We thank the reviewer for pointing this out. We note from Table 1 that increasing the number of GMM components can substantially improve posterior KL with a relatively modest latency increase; due to the quadratic complexity in number of GMM components we would expect diminishing returns here at large numbers of components, and we will make this trade-off clearer. We do not currently include a systematic sweep over transformer parameter count, so we will avoid overclaiming there and instead note it as a useful direction for future study.
>
> # Why include the prior loss? #
> The prior-loss term is not redundant with posterior supervision. As discussed in the paper, it serves two roles: it acts as a mild regularizer, and it is essential for learning a shared latent space in which the prior and posterior remain compatible for recursive reuse in sequential inference. In a small ablation on DT-5 (wide meta-prior, Exp. 4.1), removing the prior loss only slightly changes posterior Expected KL but dramatically degrades prior reconstruction Expected KL, confirming that latent-space conjugacy does not emerge automatically. We will include this ablation in the final manuscript.
> | Prior Loss? | Posterior Expected KL| Prior Expected KL|
> |-------------|--------------------------|----|
> | Yes | $0.0003 \pm 0.0000$ | $0.0004 \pm 0.0000$ |
> | No | $0.0004 \pm 0.0000$ | $0.3020 \pm 0.0190$ |

---

> > ### Author Rebuttal · Reviewer_RRZ3 · 2026-04-01
> >
> > I thank the authors for their well-formulated responses. I appreciate the proposed changes, and do not have any follow up questions. I enjoyed reading the paper (and the rebuttal), and am happy to recommend acceptance.

---

### Official Review · Reviewer_3sGq · 2026-03-12

**Soundness:** 3
**Presentation:** 4
**Significance:** 4
**Originality:** 3
**Overall Recommendation:** 5
**Confidence:** 3

**Summary:**

This work proposes the Distribution Transformer (DT), a transformer-based architecture for Bayesian inference that addresses existing challenges of Bayesian inference: 1) computational complexity, 2) inflexible prior at test time, and 3) inconsistent family structure between prior and posterior prevents sequential applications. By representing distributions as Gaussian Mixture Models (GMMs), DTs can learn an embedding of the prior, then map this embedding to the posterior using a transformer decoder without positional encodings. Finally, the posterior is unembedded. DTs are compared to multiple baseline methods for approximating both tractable and intractable posteriors. Results show that DT achieves improved posterior approximation compared to baselines with relatively low inference costs. In a sequential inference setting, similar conclusions are reached.

**Compliance With Llm Reviewing Policy:**

Affirmed.

**Final Justification:**

This paper presents a novel approach to a significant challenge, and the empirical results support the paper's claims. My original concerns related to the practical limitations of the paper, including reproducibility, scaling, and robustness. The authors' rebuttal has addressed these concerns and reinforced my prior assessment of recommending acceptance.

**Key Questions For Authors:**

1. Can the authors provide more specific details on the architectures used in each experiment setting?
2. Can the authors comment on the DT’s ability to generalize to novel meta-priors at test time?
3. How does the inference latency of DT scale with the dimension of the observation? The experiments are conducted in relatively low-dimensional domains, e.g., 4-dimensional state space in the sensor fusion setting.
4. What is the number of parameters of DT and each baseline model? Model size in MB is given, but number of parameters is the standard metric and would make for a more straightforward comparison to other models.

**Limitations:**

The authors have not discussed limitations, but this would be a beneficial inclusion.

**Strengths And Weaknesses:**

Strengths
* This paper is well-written and well-motivated. The paper’s contributions, novelty, and significance compared to prior work are made clear.
* The presented results support the authors’ claims. DT achieves strong performance with reduced latency. DT furthermore enables application to sequential domains, in which prior works are intractable or impossible.

Weaknesses
* The paper lacks some important detail for reproducing the work. While anonymized code is provided, details of the architecture are best described in the text for clarity. The general architecture design of the transformer, embedding, and unembedding modules are only provided in general terms. For example, a rule-of-thumb for the size of the embedding MLP is given, but not the exact size used for each experiment setting.
* A discussion of scaling would be a beneficial addition to the paper, due to the relevance of inference latency to the work’s motivation (see questions for more).

---

> ### Author Rebuttal · Authors · 2026-03-27
>
> We thank the reviewer for the careful reading and constructive feedback, and for recognizing the novelty, significance, and empirical strength of the work.
>
> # Architecture details #
>  We agree this should be clearer in the main paper. The supplement already provides general architecture details and several experiment-specific embedding choices; in the revision we will add a compact table in the main paper reporting, for each experiment, the prior/observation embedding sizes, number of GMM components/tokens, decoder depth/width/heads, and exact parameter counts for DT and all learned baselines.
>
> # Generalization to novel meta-priors #
> We will clarify that, like other amortized inference methods, DTs are primarily designed to generalize across the family of priors represented by the training meta-prior. Generalization to substantially out-of-support meta-priors is an important open question. To probe this, we ran a small additional ablation in Exp. 4.1 using a shifted test meta-prior with largely non-overlapping mass; both DT-5 and ACE-5 degrade, but DT remains accurate and continues to outperform ACE in this setting. We will include this ablation and phrase the conclusion carefully as evidence of some robustness to meta-prior shift in this problem, rather than broad extrapolation.
>
> Model | Training Meta-Prior Expected KL-Divergence | Unseen Meta-Prior Expected KL-Divergence
> |-------|----------------------|-------------------|
> DT-5 | $0.0003\pm 0.0000$ | $0.0054\pm 0.0015$
> ACE-5 | $0.0058\pm 0.0014$ | $0.0132\pm 0.0010$
>
> # Latency scaling #
> We thank the reviewer for highlighting this. We do not currently include a dedicated sweep over observation dimensionality, and we will clarify this limitation. In DT, raw observations are first embedded, so compute depends more directly on the number of observation tokens and the latent/GMM token sizes than on raw observation dimension itself. Empirically, Appendix C shows favourable scaling with latent-state dimension, and the sequential experiments already include a challenging setting with 30 observations per step. We will add discussion of this in the revised manuscript.
>
> # Parameter counts #
> We agree that parameter counts are more standard than model size in MB, and we will report exact counts for all learned models in the revision.
>
> # Limitations #
> We agree a dedicated limitations paragraph would improve the paper. We will add discussion of (i) dependence on the training meta-prior family, (ii) approximation limits of finite-component GMMs, (iii) computational scaling with mixture size and latent dimension, and (iv) the need for more systematic study of out-of-support meta-prior generalization.
> We thank the reviewer again---these suggestions will directly improve the clarity and reproducibility of the paper.

---

> > ### Author Rebuttal · Reviewer_3sGq · 2026-04-02
> >
> > Thank you for the comprehensive response. My concerns have been fully addressed.

---

### Decision · Program_Chairs · 2026-04-30

**Decision:**

Accept (spotlight)

**Comment:**

This paper tackles a highly relevant bottleneck in amortized Bayesian inference: the computational expense of existing approximate inference methods and the necessity of retraining when priors change, which severely limits their utility in sequential tasks like real-time sensor fusion. To address this, the authors introduce the Distribution Transformer (DT), an architecture that represents both prior and posterior distributions as tokenized GMMs. By learning a mapping within the same parametric family, the model achieves an "approximately conjugate" inference, enabling the posterior to be seamlessly reused as the next step's prior in recursive filtering tasks.

There is a strong, unanimous consensus among the reviewers in favor of accepting this paper. The reviewers highlighted four core strengths that make this an excellent contribution:
* **Strong Motivation:** The necessity for prior-flexible, amortized inference that can handle sequential composition is an underexplored but critical challenge for practitioners.
* **Novel Approach:** The method of tokenizing distributions as unordered GMM components and processing them without positional encodings to maintain permutation equivariance is both original and elegant. Reviewers also noted the strong theoretical grounding of the training objective.
* **Empirical Performance:** The paper is backed by robust experimental validation. The results clearly demonstrate that DTs can achieve fast inference times, consistently matching or outperforming established baselines (SVI, PFNs, TabPFNv2, and ACE) across static and sequential tasks.
* **High-Quality Presentation:** Reviewers universally praised the manuscript for its exceptional clarity and readability.

By providing compelling new ablations that addressed concerns about recursive error accumulation, generalization across GMM component counts, and robustness to meta-prior shifts, the authors actively and successfully resolved all major reviewer concerns.

**Additional feedback for the camera-ready revision:**
While the paper is technically sound, well-written, and undeniably useful to the ICML community, I encourage the authors to also briefly discuss the following aspects:
* **Training Time and Cost Trade-offs:** OIffline training times are substantially higher than some baselines. For example, in some experiments DT required substantially longer to train than PFNs. A brief note acknowledging the upfront computational cost required to achieve this test-time speed would be valuable.
* **Memory Constraints and Scalability:** In the higher-dimensional Factor-Structure Stochastic Volatility study, the authors note that extreme memory requirements necessitated substituting the Gaussian KDE for a moment-matched multivariate Gaussian and prohibited parallel series processing. Furthermore, this experiment required smaller batch sizes to run. Openly discussing these memory bottlenecks will help practitioners scale the model effectively.
* **Hyperparameter Sensitivity:** While Appendix D details a fixed transformer architecture, Table 5 notes that hyperparameters were obtained via "limited manual tuning, adjusted until training was stable in all cases". Commenting on the model's sensitivity to these specific hyperparameter choices would further strengthen the paper's reproducibility.

Overall, this is an excellent, original piece of research that I believe should clearly be accepted to the conference.